# Differences in local immune cell landscape between Q fever and atherosclerotic abdominal aortic aneurysms identified by multiplex immunohistochemistry

**Kimberley RG Cortenbach[1†], Alexander HJ Staal[1†], Teske Schoffelen[2], Mark AJ Gorris[1], Lieke L Van der Woude[1], Anne FM Jansen[2], Paul Poyck[3], Robert Jan Van Suylen[4], Peter C Wever[5], Chantal P Bleeker-Rovers[2], Mangala Srinivas[1,6], Konnie M Hebeda[7], Marcel van Deuren[2], Jos W Van der Meer[2], Jolanda M De Vries[1], Roland RJ Van Kimmenade[8]***

[1]Department of Tumor Immunology, Radboud Institute for Molecular Life Sciences, Radboud University Medical Center, Nijmegen, Netherlands; [2]Department of Internal Medicine, Division of Infectious Diseases and Radboud Center for Infectious Diseases, Radboud University Medical Centre, Nijmegen, Netherlands; [3]Department of Surgery, Radboud University Medical Centre, Nijmegen, Netherlands; [4]Department of Pathology, Jeroen Bosch Ziekenhuis, 's Hertogenbosch, Netherlands; [5]Department of Medical Microbiology and Infection Control, Jeroen Bosch Ziekenhuis, 's Hertogenbosch, Netherlands; [6]Department of Cell Biology and Immunology, Wageningen University, Wageningen, Netherlands; [7]Department of Pathology, Radboud University Medical Centre, Nijmegen, Netherlands; [8]Department of Cardiology, Radboud University Medical Centre, Nijmegen, Netherlands

**\*For correspondence:**
Roland.vanKimmenade@
radboudumc.nl

†These authors contributed
equally to this work

## Abstract

**Background:** Chronic Q fever is a zoonosis caused by the bacterium *Coxiella burnetii* which can manifest as infection of an abdominal aortic aneurysm (AAA). Antibiotic therapy often fails, resulting in severe morbidity and high mortality. Whereas previous studies have focused on inflammatory processes in blood, the aim of this study was to investigate local inflammation in aortic tissue.

**Methods:** Multiplex immunohistochemistry was used to investigate local inflammation in Q fever AAAs compared to atherosclerotic AAAs in aorta tissue specimen. Two six-plex panels were used to study both the innate and adaptive immune systems.

**Results:** Q fever AAAs and atherosclerotic AAAs contained similar numbers of CD68+ macrophages and CD3+ T cells. However, in Q fever AAAs, the number of CD68+CD206+ M2 macrophages was increased, while expression of GM-CSF was decreased compared to atherosclerotic AAAs. Furthermore, Q fever AAAs showed an increase in both the number of CD8+ cytotoxic T cells and CD3+CD8-FoxP3+ regulatory T cells. Finally, Q fever AAAs did not contain any well-defined granulomas.

**Conclusions:** These findings demonstrate that despite the presence of pro-inflammatory effector cells, persistent local infection with *C. burnetii* is associated with an immune-suppressed microenvironment.

**Funding:** This work was supported by SCAN consortium: European Research Area - CardioVascualar Diseases (ERA-CVD) grant [JTC2017-044] and TTW-NWO open technology grant [STW-14716].

## Editor's evaluation

This is a collaborative study of clinical centers that investigates tissue pathology and immune cell infiltration of aortic aneurysms from chronic Q fever patients. The combination of precious and rare human tissue samples with well-designed multiplex IHC panels for characterizing local immune responses within the spatial context is unique in the field of human infection immunology and has revealed unprecedented insight into the manifestation of this disease.

## Introduction

Q fever is a zoonosis caused by the Gram-negative intracellular bacterium *Coxiella burnetii (C. burnetii)*, with natural reservoirs in a wide range of wild and domestic animals. In infected animals (such as goats) milk, placenta, and birth fluids, can contain this microorganism, which may cause human infections via inhalation. Acute Q fever can present as pneumonia, hepatitis, and isolated fever, yet 60% of cases are asymptomatic (*Maurin and Raoult, 1999*). Progression to chronic Q fever occurs in approximately 5% of infected individuals *Fournier et al., 1998*; people at risk are older, suffer from aortic or iliac aneurysm, or renal insufficiency (*Kampschreur et al., 2012*), or previously underwent valvular or vascular prosthesis surgery (*Kampschreur et al., 2012*). Chronic Q fever manifests as endocarditis or vascular Q fever, that is, infection of an abdominal aortic aneurysm (AAA) or vascular prosthesis (*Wegdam-Blans et al., 2011*; *Botelho-Nevers et al., 2007*; *van Roeden et al., 2019*).

Vascular manifestations of Q fever can have severe clinical consequences. In a population of proven and probable vascular manifestations of Q fever patients according to the Dutch consensus guideline, a Dutch cohort study has described that complications had occurred in 61% of the cases. Of these, acute complications (i.e., rupture, dissection, endoleak, or symptomatic aneurysms) were most prevalent (35%), followed by abscesses (22%), and fistula (14%). Moreover, 25% of patients had a definitely or probably chronic Q fever related cause of death (*van Roeden et al., 2019*). In addition, serological screening of 770 patients with aorto-iliac disease, for example, aneurysms or previous vascular reconstructions, demonstrated that 16.9% was seropositive for Q fever, of which 30.8% suffered from chronic Q fever. In this group, aneurysm-related acute complications were more common than in aneurysm patients without Q fever (*Hagenaars et al., 2014b*).

To elucidate the pathology underlying chronic Q fever, previous studies have mainly focussed on immune responses in peripheral blood. Blood mononuclear cells of patients with chronic Q fever, when exposed to *C. burnetii* in vitro, produce high amounts of Interferon-gamma (IFNg), the proinflammatory cytokine considered crucial for killing of the pathogen (*Schoffelen et al., 2014*; *Schoffelen et al., 2017*). In the infected tissues, *C. burnetii* resides and replicates in monocytes and macrophages (*Ghigo et al., 2009*). In vascular manifestations of Q fever, it is assumed that *C. burnetii* survives in resident macrophages in the vascular wall (*Lepidi et al., 2009*; *Lepidi et al., 2003*). In such patients, there is an apparent inability to effectively eradicate *C. burnetii*, despite the aforementioned IFNg response. In general, a pro-inflammatory response with granuloma formation and intracellular killing or control of the bacterium by activated 'M1' monocytes/macrophages is required to contain intracellular infections like Q fever. Surprisingly, the apparent inability of chronic Q fever patients to kill *C. burnetii* has only been investigated in studies in vitro which showed suggestive roles for polarization to tolerogenic M2 macrophages and increased numbers of circulating regulatory T cells (*Layez et al., 2012*; *Benoit et al., 2008*).

It is still unclear how *C. burnetii* survives and locally escapes the immune system in vascular manifestations of Q fever. To address this, we have investigated the local immune response in *C. burnetii*-infected AAAs (Q fever AAA), classical atherosclerotic AAA, acutely infected AAA, and control aorta tissue, applying multiplex immunohistochemistry (mIHC) on human patient tissues. We investigated both the adaptive and innate immune systems. We show that in vascular manifestations of Q fever numerous immune suppressive mechanisms appear to be present, including the absence of pro-inflammatory granulomas, increased numbers of regulatory T cells, polarization of macrophages into the tolerogenic M2 phenotype, and decreased expression of GM-CSF.

# Materials and methods

Abdominal aorta tissue samples from patients with Q fever infected aneurysms and control groups (i.e., atherosclerotic AAAs, acutely infected AAAs, and non-aneurysmatic aortas) were investigated with a novel mIHC method to study the involvement of the innate and adaptive immune system in vascular manifestations of Q fever. The data underlying this article will be shared upon reasonable request to the corresponding author.

## Patient samples

Tissue samples were collected from four groups of patients in two Dutch hospitals: Jeroen Bosch Hospitals in 's-Hertogenbosch and Radboud University Medical Center in Nijmegen. The first group consisted of patients diagnosed with *C. burnetii* infected AAA (Q fever AAA) according to the Dutch consensus guideline (*Kampschreur et al., 2015*): all patients had an abdominal aneurysm (AAA) and IgG phase I was at least 1:1024 in combination with a positive PCR of aortic tissue. The second group consisted of patients with atherosclerotic AAA without clinical suspicion of Q fever which were selected from our database at random. The third group consisted of patients with an acutely infected AAA, with the same definition of AAA in combination with positive cultures of *Streptococcus pneumoniae* and *Streptococcus Agalactiae*, respectively. In these three groups, AAA was defined as a CT-proven AAA with a diameter of at least 3.0 cm (*Johnston et al., 1991*). All aneurysmatic tissue samples were either obtained from patients undergoing elective surgical repair or emergency repair in case of aortic rupture. The fourth group consisted of abdominal aorta samples from patients undergoing kidney explantation surgery for transplantation purposes, with an aortic diameter smaller than 3.0 cm. Due to the limited availability of acutely infected AAAs and non-aneurysmatic aortas, all available and eligible samples were selected. The samples from Jeroen Bosch Hospital were described in a previous study. (*Hagenaars et al., 2014a*).

**Table 1.** Overview of the used markers and clones per panel, including definition of each cell type as used for our analysis.

| | Adaptive immune system | | Innate immune system | |
|---|---|---|---|---|
| Markers (clone) | DAPI | | DAPI | |
| | CD3 (SP7) | | CD68 (PG-M1) | |
| | CD8 (CD8/144B) | | CD206 (CL038+) | |
| | CD20 (L26) | | CD15 (MMA) | |
| | CD1c (2F4) | | CD31 (JC70A) | |
| | FoxP3 (236A/E7) | | MMP9 (polyclonal) | |
| | CD45RO (UCHL-1) | | GM-CSF (polyclonal) | |
| | Autofluorescence | | Autofluorescence | |
| | | | | |
| Cell phenotype | T cell | CD3+ | Macrophage | CD68+ |
| | Helper T cell | CD3+ CD8− | M1-like macrophage | CD68+ CD206− |
| | Cytotoxic T cell | CD3+ CD8+ | M2-like macrophage | CD68+ CD206+ |
| | Regulatory T cell | CD3+ CD8− FoxP3+ | Neutrophil | CD15+ |
| | Memory T cell | CD3+ CD45RO+ | Endothelium | CD31+ |
| | B cell | CD20+ | MMP9+ cell | MMP9+ CD15− |
| | Classic DC type 2 | CD1c+ CD20− | MMP9+ neutrophil | MMP9+ CD15+ |
| | DAPI | Nucleus | DAPI | Nucleus |
| | Autofluorescence | Elastin fibers | Autofluorescence | Elastin fibers |

The medical ethics committees of the institutions approved the study, in line with the principles outlined in the Declaration of Helsinki (Radboudumc: 2017-3196; Jeroen Bosch Hospital: 2019.05.02.01).

## Tissue processing

During surgery, the ventral part of the abdominal aorta was removed. If necessary, adhering thrombi were gently removed from the tissue before further processing. Directly after collection, samples were fixed in buffered 4% formaldehyde for at least 24 hr and no longer than 72 hr. If large amounts of calcification were present, samples were decalcified by storing them in EDTA solution for another 24 hr. Subsequently, samples were carefully embedded in paraffin in an attempt to include all aorta layers (formalin-fixed and paraffin-embedded [FFPE]). Of these tissues, full-thickness transverse sections of 4 μm were mounted on silane-coated glass slides (New Silane III, MUTO PURE CHEMICALS, Japan).

## Multiplex immunohistochemistry

Samples were stained with two mIHC panels, which enclosed the innate and adaptive immune system (*Table 1*). Optimization and validation of mIHC panels were performed as described previously (*Gorris et al., 2018*). Samples were stained with six consecutive tyramide signal amplification (TSA) stains followed by antigen stripping after every staining. This resulted in the fluorophore remaining on the target, thus enabling eight simultaneous colors on one slide (six markers, DAPI, and autofluorescence). Slides were stained automatically in a Leica Bond system (BOND-Rx Fully Automated IHC and ISH, Leica Biosystems). After positioning in the machine, slides were deparaffinized, rehydrated, and washed with demi water. After this, samples underwent heat-induced antigen retrieval (HIER) in BOND Epitope Retrieval 2 (AR9640, Leica Biosystems) or BOND Epitope Retrieval 1 (AR9961, Leica Biosystems) for 20 min for the adaptive and innate panel, respectively. Then, protein blocking in Akoya Antibody Diluent/Block (Akoya Biosciences, MA) took place for 10 min, followed by incubation with the first primary antibody for 1 hr, subsequently with the secondary antibody (Polymer HRP, Ms+ Rb [Akoya Biosciences, MA]) for 30 min and finally with an Opal fluorophore (Akoya Biosciences, MA) dissolved 1:50 in 1× Plus Amplification Diluent (Akoya Biosciences, MA) for 10 min. To facilitate multiplex staining with six markers, samples were heated for 10 min which enabled antigen stripping. After this staining cycle, this procedure was repeated for five different primary antibodies, the secondary antibody, and corresponding Opal fluorophores. Finally, DAPI was used as a nuclear counterstain and slides were mounted with Fluoromount-G (0100-01; Southern Biotech, Birmingham, AL). All incubations steps were performed at room temperature. Please see *Supplementary file 1B* for a more detailed overview of the used reagents.

## Imaging, multispectral unmixing, and analysis

After staining, image acquisition and immune cell quantification were performed using an automated approach. First, the PerkinElmer Vectra (Vectra 3.0.3; PerkinElmer, MA) scanned whole slides at 4× magnification and 20× magnification, allowing precise cell segmentation incorporating the entire sample. The average 20× views per slide was 283 resulting in an average tissue area of 59.0±32.6 mm$^2$, and this high number substantially reduces the chance of sampling bias. Spectral libraries and inForm Advanced Image Analysis software (inForm 2.4.8; Akoya Biosciences, MA) unmixed these multispectral images (*Figures 1A and 2A*).

Subsequently, inForm Advance Imaging Analysis software was used for segmentation of tissue and cells. For tissue segmentation, tissue slides were divided into tissue, infiltrate, thrombus, blood, and background (*Figure 1C*). This segmentation was based on DAPI, autofluorescence and, if present, also CD20 and CD3. For single-cell segmentation, cells were identified with DAPI and autofluorescence, and, depending on the panel, with membrane markers CD20 and CD3 (*Figure 1B*) or CD68, CD206, and CD15. Requiring DAPI for cell segmentation ensured the exclusion of artifact staining, in which DAPI is absent. The output of the software was 20× magnification images and cell data (localization, tissue, phenotype, and marker) per slide. Images were combined into single flow cytometry standard (fcs) files, allowing analysis in FlowJo (FlowJo 10.0.7, Becton Dickinson, NJ). In FlowJo, only cells in tissue and infiltrate were analyzed and gates were drawn as shown in *Figure 1E* for the adaptive panel and *Figure 2B* for the innate panel by two observers with excellent interobserver correlation (*Figures 1D and 2D*).

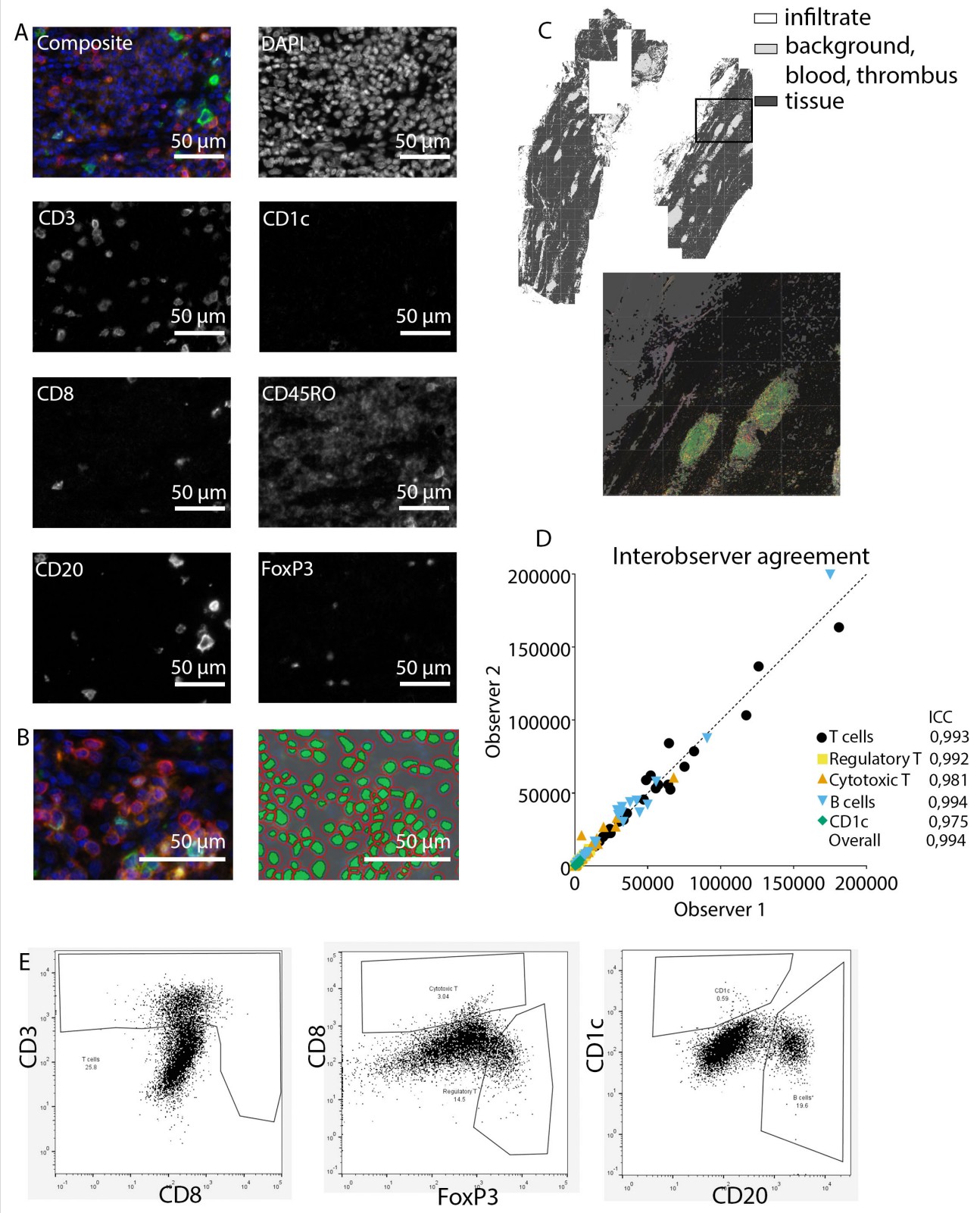

**Figure 1.** Analysis of adaptive immune system panel. (**A**) Composite image and separate channels. (**B**) Composite image and cell segmentation tool. (**C**) Tissue segmentation tool (upper image) and overlay with staining, which shows overlay in segmented and actual stained infiltrate. (**D**) Interobserver agreement with high intraclass coefficients, supporting our robust method. (**E**) Representative FACS plots for drawing cell populations.

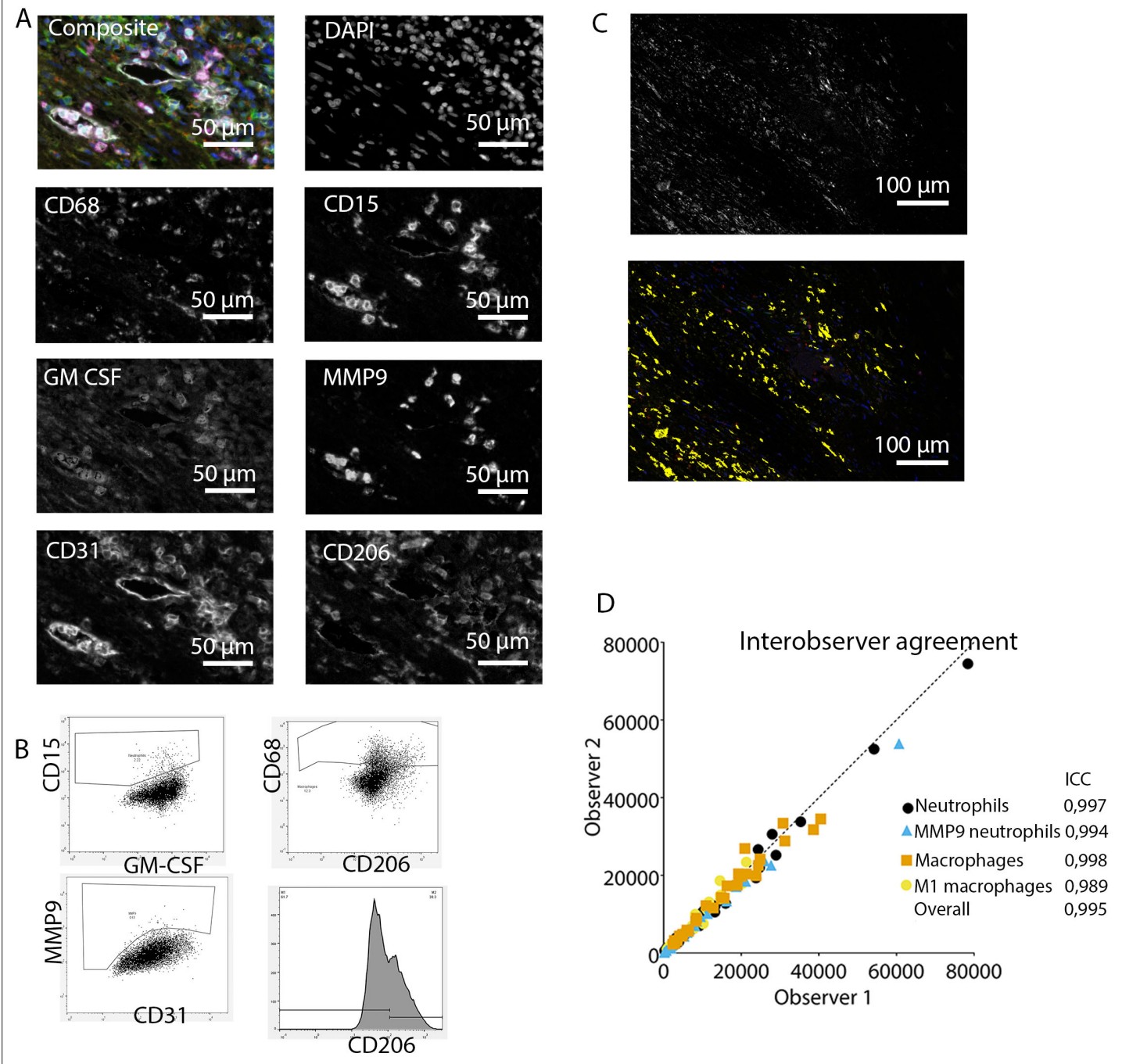

**Figure 2.** Analysis of innate immune system panel. (**A**) Composite image and separate channels. (**B**) Representative FACS plots for drawing cell populations. (**C**) inForm threshold analysis of GM-CSF expression. Upper image: GM-CSF channel. Lower image: GM-CSF signal above threshold. (**D**) Interobserver agreement with high intraclass coefficients.

These clear distinct positive cell populations were not found for CD45RO and MMP9 as their expression is gradual (*Booth et al., 2010*). For that matter, the gates for these markers were drawn in negative populations, namely non-T cells and non-neutrophils, respectively. Following this, these gates were copied to populations that could express these markers. GM-CSF fcs files did not show distinct positive and negative populations although they were visible in the microscopy images. Therefore, inForm Advance Imaging Analysis software was used for automatic thresholding for GM-CSF per sample, providing the number of GM-CSF positive pixels per sample (*Figure 2C*).

## Histology

To study vessel wall architecture, tissue samples were stained with hematoxylin-eosin (HE) and Elastin von Gieson (EVG).

## Quantification of tertiary lymphoid structures

Based on HE staining, a TLS was defined as a nodular structure consisting of at least 50 B and T lymphocytes. These structures are fluent and in case of constriction, split in multiple TLS.

## Statistical analysis

SPSS for Windows (IBM Corp, 2017. IBM SPSS Statistics for Windows, Version 25.0. Armonk, NY: IBM Corp) was used for statistical analysis. PRISM 8.0.2 (Graphpad, GSL Biotech LLC, CA) was used for visualization of results. Continuous data were expressed as mean ± standard deviation (SD), or in case of non-Gaussian distribution, as median (interquartile range; IQR). Kruskal-Wallis test adjusted with Bonferroni correction for multiple testing was used for testing continuous variables between four groups. Binary variables were tested for differences using the Fisher exact test. Interobserver variability was calculated with the intraclass correlation coefficient. Correlations between continuous non-Gaussian distributed variables were studied with Kendall's tau because of low numbers per group. $p < 0.05$ was considered statistically significant. Principal component analysis was performed in RStudio 1.2.5033 (RStudio, Inc Boston, MA) and R (R Foundation for Statistical Computing, Vienna, Austria) using singular value decomposition and the tidyverse (*Wickham et al., 2019*) and factoextra packages (*Alboukadel Kassambara, 2020*).

# Results

## Baseline characteristics

This study includes 10 Q fever AAAs (with *C. burnetii* PCR positive on aortic tissue), 12 classical atherosclerotic AAAs, 2 acutely infected AAAs (with positive cultures of *Streptococcus* species), and 5 normal abdominal aorta tissues. *Table 2* presents baseline characteristics of the cohort, showing that the majority of patients in all groups were males of older age (median 71 years old in Q fever group), and cardiovascular risk factors were common, including hypertension, diabetes mellitus, hypercholesterolemia, and smoking. Additionally, the aortic diameter was similar amongst the groups.

## Immune cell activation in Q fever, atherosclerotic, and acutely infected AAAs compared to normal aortas

Our mIHC technique reveals activation of both the innate and the adaptive immune systems in Q fever, atherosclerotic, and acutely infected AAAs compared to normal abdominal aortas (*Figure 3*). In contrast to normal abdominal aortas, all Q fever AAAs, atherosclerotic AAAs, and acutely infected AAAs showed impressive lymphocyte accumulation and proliferation with very large tertiary lymphoid structures (TLS) present in the adventitial layer (*Figure 3F, J and N*) (20% vs. 100% vs. 100% vs. 100%, respectively; p=0.000). Importantly, well-defined granulomas were neither observed in any of the Q fever AAAs, nor in the other groups. In Q fever AAA, elevated numbers of CD3[+] T cells (p=0.010) and CD20[+] B cells (p=0.012) were observed compared to normal aortas. Atherosclerotic AAAs revealed increased numbers of CD3[+] T cells (p=0.005), CD20[+] B cells (p=0.003), and CD15[+] neutrophils (p=0.023) compared to control. There were no significant differences in the numbers of cells between Q fever AAA and atherosclerotic AAA. As expected, acutely infected AAAs showed an increase in neutrophils compared to normal aortas (p=0.026). Numbers of CD1c[+] classical dendritic cell type 2 (cDC2) and CD68[+] macrophages were similar among all groups.

The principal component analysis (PCA) demonstrated when using these markers, a clear distinct population with normal aorta samples is formed, whilst the atherosclerotic and Q fever population completely overlapped (*Figure 4A*). Therefore, we aimed to investigate which markers differ between these two groups. If we add cell subset markers for macrophages and T cells, this overlap has completely disappeared, indicating these subset markers differentiate these groups (*Figure 4B*). Below, we will elucidate the differences in cell subsets between atherosclerotic AAA and Q fever AAA.

**Table 2.** Baseline characteristics.

| Characteristic | Normal (N=5) | Missing data normal | Atherosclerotic AAA (N=12) | Missing data atherosclerotic | Q fever AAA (N=10) | Missing data Q fever | Infectious AAA (N=2) | Missing data infectious | Significance |
|---|---|---|---|---|---|---|---|---|---|
| Male | 3 (60%) | 2 | 10 (83.3%) | 0 | 9 (90.0%) | 1 | 1 (50.0%) | 1 | 0.482 |
| Age | 65 (54–68) | 0 | 72 (66–78) | 0 | 71 (64–77) | 0 | 67 (62–) | 0 | 0.307 |
| Length | 1.80 (1.69–1.83) | 0 | 1.77 (1.69–1.80) | 0 | 1.78 (1.70–) | 8 | 1.76–(1.76–1.76) | 1 | 0.923 |
| Weight | 80.0 (77.5–102.5) | 0 | 82.6 (75.5–92.6) | 0 | 88.5 (79–) | 8 | 94.8 (94.8–94.8) | 1 | 0.637 |
| BMI | 27.0 (22.1–34.7) | 0 | 26.8 (24.0–29.4) | 0 | 27.8 (27.3–) | 8 | 30.6 (30.6–30.6) | 1 | 0.651 |
| Hypertension | 3 (60%) | 0 | 8 (66.7%) | 0 | 6 (60.0%) | 2 | 1 (50.0%) | 1 | 1.000 |
| Hypercholesterolemia | | 5 | 9 (75.0%) | 0 | 6 (60.0%) | 2 | 0 (0.0%) | 1 | 0.426 |
| DM | 1 (20%) | 0 | 3 (25.0%) | 0 | 1 (10.0%) | 2 | 0 (0.0%) | 1 | 0.853 |
| Total cholesterol | | 5 | 3.6 (3.0–5.2) | 5 | 4.7 (3.2–6.1) | 5 | | 2 | 0.371 |
| HDL | | 5 | 0.9 (0.8–1.1) | 6 | 0.9 (0.9–1.3) | 5 | | 2 | 0.583 |
| Previous aorta surgery | 0 (0.0%) | 0 | 6 (50.0%) | 0 | 0 (0.0%) | 0 | 0 (0.0%) | 1 | 0.034* |
| Rupture | 0 (0.0%) | 0 | 0 (0.0%) | 0 | 3 (30.0%) | 0 | 0 (0.0%) | 0 | 0.193 |
| Artery disease | 1 (20%) | 4 | 11 (91.7%) | 0 | 4 (40.0%) | 2 | 1 (50.0%) | 1 | 0.002* |
| Smoking | 3 (60%) | 5 | 11 (91.7%) | 1 | 6 (60.0%) | 1 | 1 (50.0%) | 1 | 0.387 |
| Packyears | 27 (27–27) | 4 | 33 (20–50) | 2 | 45 (0–53) | 1 | 0 (0–0) | 1 | 0.819 |
| CRP | | 5 | | 12 | 12.5 (12.0–65) | 4 | | 2 | n/a |
| Diameter CT | | 5 | 57 (55–71) | 0 | 60 (45–81) | 1 | 46 (46–46) | 1 | 0.397 |
| Beta blocker | 1 (20%) | 3 | 8 (66.7%) | 0 | 3 (30.0%) | 2 | 0 (0.0%) | 1 | 0.567 |
| ARB/ACEi | 2 (40%) | 2 | 6 (50.0%) | 0 | 3 (30.0%) | 2 | 0 (0.0%) | 1 | 0.789 |
| Calciumblocker | 0 (0%) | 3 | 3 (25.0%) | 0 | 0 (0.0%) | 2 | 0 (0.0%) | 1 | 0.439 |
| Diuretics | 0 (0%) | 4 | 1 (8.3%) | 0 | 2 (20.0%) | 0 | 1 (50.0%) | 1 | 0.009* |

* Represents p≤0.05. Numbers display numbers of patients with percentage or median with interquartile range (IQR).

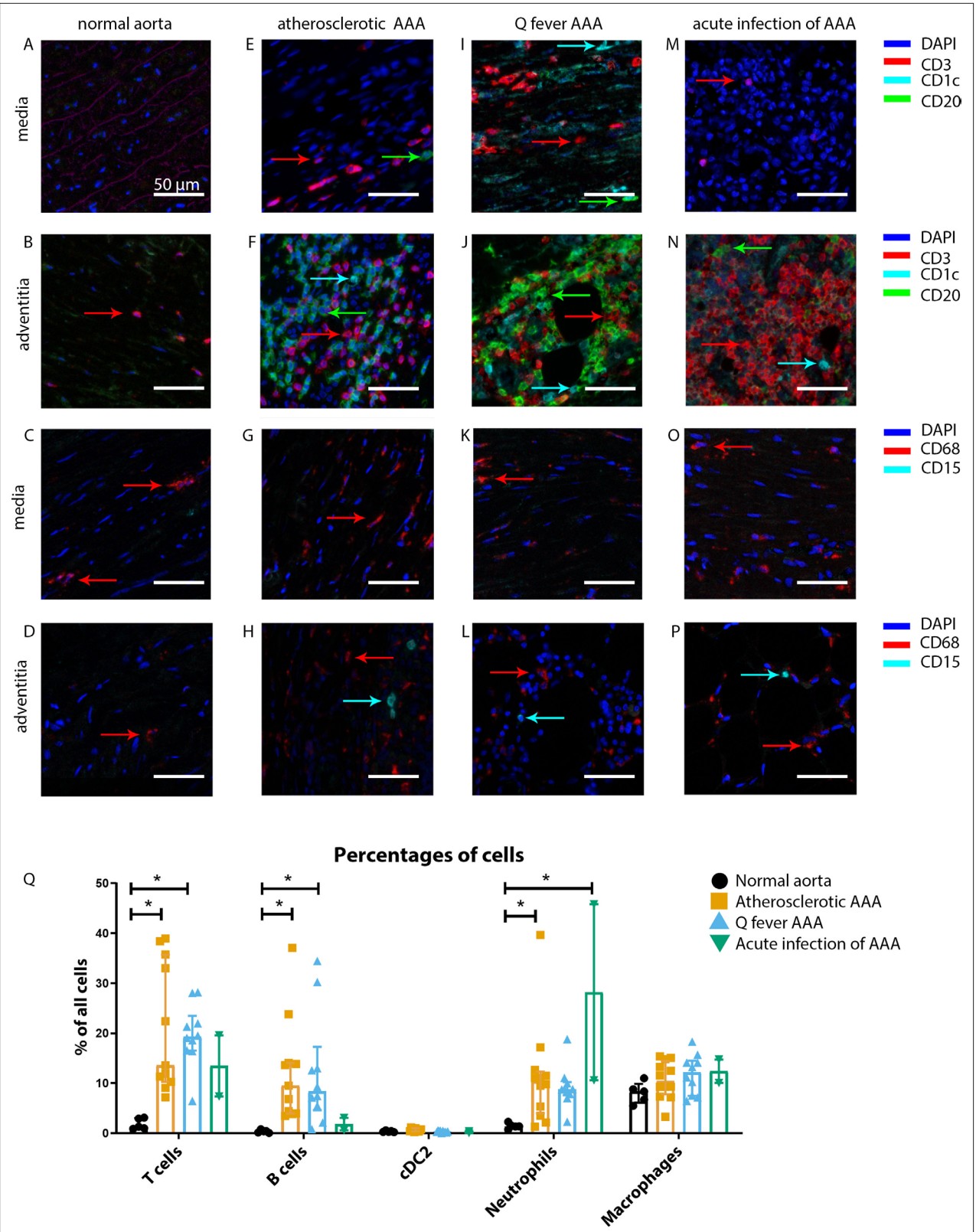

**Figure 3.** Immune cell activation in atherosclerotic, Q fever infected, and acutely infected AAAs. All scale bars represent 50 µm. (**A–P**): Adaptive (**A, B, E, F, I, J, M, N**) and innate (**C, D, G, H, K, L, O, P**) immune cells in a representative normal abdominal aorta, atherosclerotic AAA, Q fever AAA, and acutely infected AAA. Arrows with corresponding colors indicate the presence of immune cells with red for CD3+ T cells, cyan for CD1c+ cDC2, and green for CD20+ B cells in the adaptive panel (**A, B, E, F, I, J, M, N**); and red for CD68+ macrophages and cyan for CD15+ neutrophils in the innate panel

*Figure 3 continued on next page*

*Figure 3 continued*

(**C, D, G, H, K, L, O, P**). (**Q**): quantification of percentages of different types of immune cells in the whole tissue sections, showing the increases in T and B cells in atherosclerotic AAA and Q fever AAA compared to normal and increase in neutrophils in acute infection and atherosclerotic AAA compared to normal. Note that there are no differences between atherosclerotic AAA and Q fever AAA. * Represents p≤0.05. Source data can be found in *Figure 3—source data 1*.

The online version of this article includes the following source data for figure 3:

**Source data 1.** Immune cell activation in atherosclerotic, Q fever infected, and acutely infected AAAs.

## Q fever AAAs show a shift toward M2 macrophages

To investigate whether there are differences in innate immune system activation between atherosclerotic AAA and Q fever AAA, we used mIHC for description of macrophage subset populations based on CD68+CD206− (M1 macrophages) and CD68+CD206+ (M2 macrophages) and for expression of matrix metalloproteinase-9 (MMP9) and Granulocyte Macrophage Colony Stimulating Factor (GM-CSF). As demonstrated in *Figure 5*, CD206 expression colocalized with CD68 in M2 macrophages. We found that in Q fever AAAs, the number of CD206+ M2 macrophages was higher than in atherosclerotic aortas (p=0.005) (*Figure 5J*). These aortas also revealed lower levels of the pro-inflammatory cytokine GM-CSF when corrected for the percentage of macrophages and CD206+ M2 macrophages (p=0.033 and p=0.007, respectively) (*Figure 6*). On the other hand, atherosclerotic AAAs showed a larger amount of the more pro-inflammatory CD206− M1 subset compared to Q fever AAAs (p=0.005) (*Figure 5J*), combined with a higher expression of GM-CSF per macrophage (p=0.033) (*Figure 6E*). Additionally, we observed a larger MMP9+ proportion of macrophages in atherosclerotic AAAs than in Q fever AAAs (p=0.04). These findings are compatible with extensive chronic inflammation but an immune-suppressed environment in Q fever AAAs in contrast to atherosclerotic AAAs.

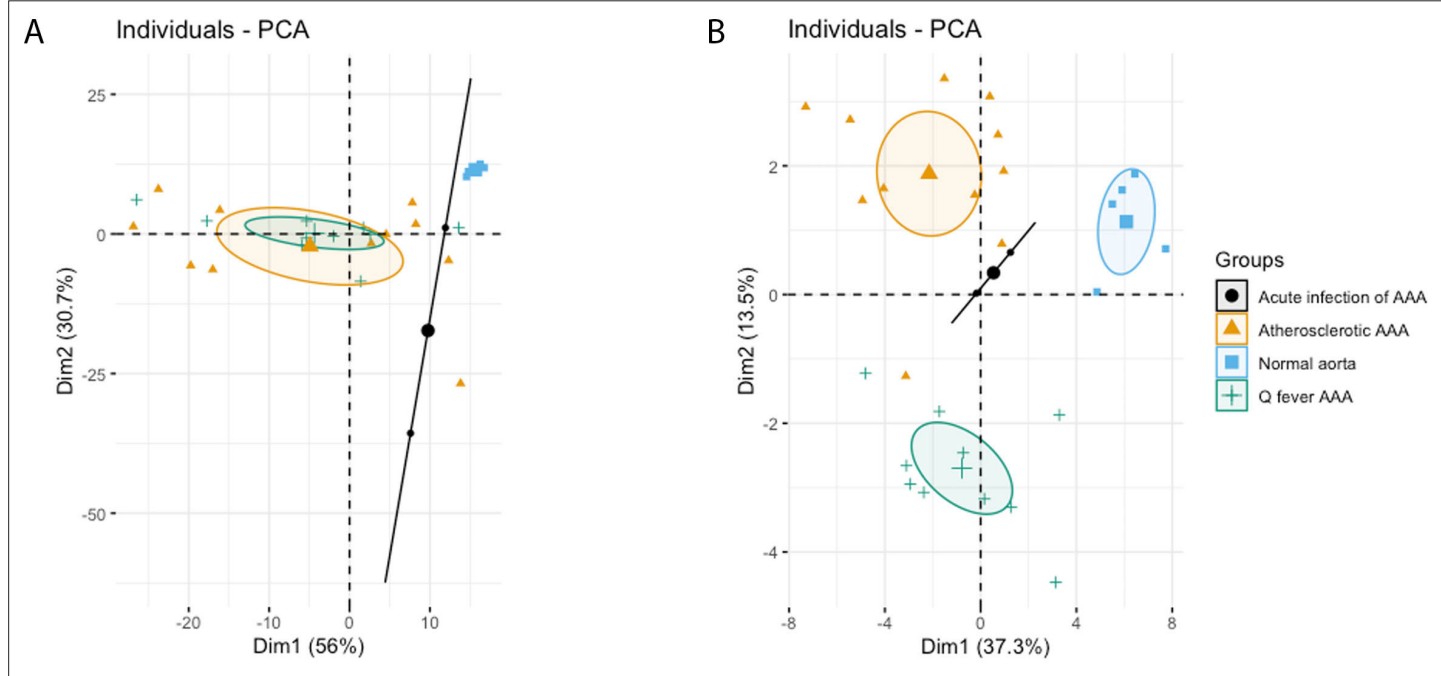

**Figure 4.** Principal Component Analyses. (**A**) Principal component analysis (PCA) including CD3, CD20, CD68, CD15, and CD1c. There is a clear distinct population consisting of normal abdominal aortas. There are two data points for acute infection, resulting in a line. Intriguingly, atherosclerotic AAA and Q fever infected AAA are completely overlapping. This indicates that these populations are similar when testing for these cell markers. (**B**): PCA including all markers (CD68, CD15, MMP9, GMCSF, CD31, CD206, CD3, CD1c, CD8, FoxP3, CD45RO, and CD20). Note the difference with (**A**): here all groups form separate populations, indicating that the newly added markers including subset markers describe the differences between atherosclerotic and Q fever AAA. See *Supplementary file 1A* for loadings of both PCAs.

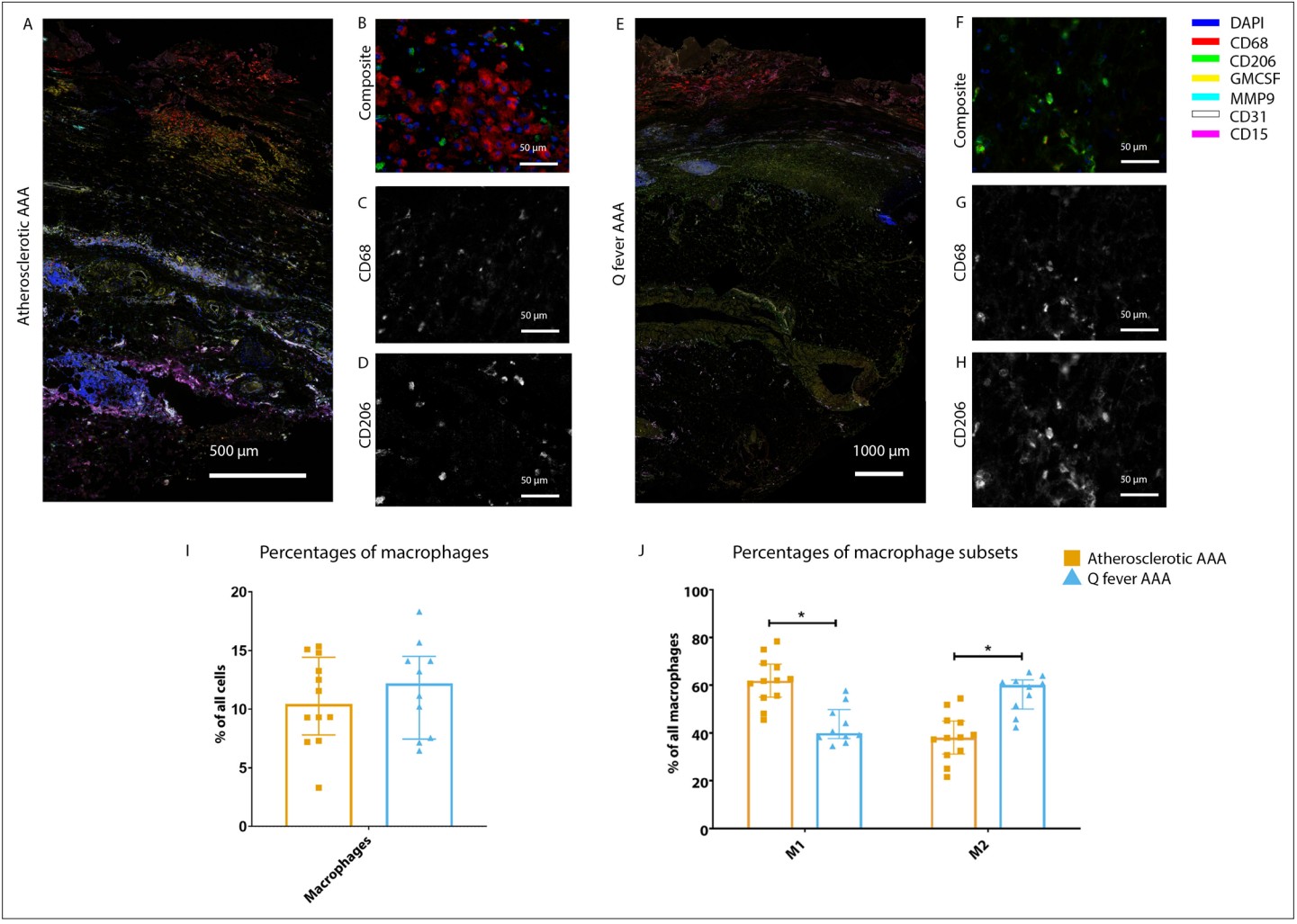

**Figure 5.** Phenotype shift in macrophages in Q fever AAA towards M2. (**A**): Overview photo of atherosclerotic AAA, upper portion is intima layer, lower portion adventitia. (**B**): Composite of CD68 and CD206 with majority CD68. (**C, D**): Separated channels for CD68 and CD206, respectively. (**E**): Overview of Q fever infected AAA, with the same orientation as (**A**). (**F**): Composite of CD68 and CD206, with mostly CD206+ cells which also express CD68, as supported by separated channels in (**G**) and (**H**). (**I, J**): Quantification of percentages of macrophages in entire tissue sections (**I**) and of proportions of M1 and M2 macrophages in these macrophages (**J**), showing the phenotype switch in Q fever AAAs toward M2. * Represents p≤0.05. Source data can be found in *Figure 5—source data 1*.

The online version of this article includes the following source data for figure 5:

**Source data 1.** Phenotype shift in macrophages in Q fever AAA towards M2.

## Increased cytotoxic T cells and regulatory T cells in Q fever AAAs

To study the involvement of the adaptive immune system in both groups, tissues were stained with antibodies against CD8 for cytotoxic T cells, FoxP3 for regulatory T cells, and CD45RO for memory T cells. Helper T cells were defined as CD3+ T cells without CD8 expression.

Whereas the number of CD3+ T cells is equal among vascular manifestations of Q fever and atherosclerotic AAA samples, samples from patients with vascular manifestations of Q fever exhibited larger numbers of CD3+CD8+ cytotoxic T cells (p=0.000), coinciding with a decrease of CD3+CD8− helper T cells (p=0.000) and CD3+CD45RO+ memory T cells (p=0.023)(*Figure 7G*). Infiltrates were formed, as depicted by the strong correlation between T- and B-cells (correlation coefficient 0.6 [CI 0.404–0.788], p=0.000).

When correcting numbers of T cell subsets for infiltrate area, we found an increase in the number of CD3+CD8+ cytotoxic T cells in Q fever AAA compared to atherosclerotic AAA (p=0.013) (*Figure 7H*). If the number of T cell subsets is calculated per mm2 tissue (defined as entire sample area minus infiltrate

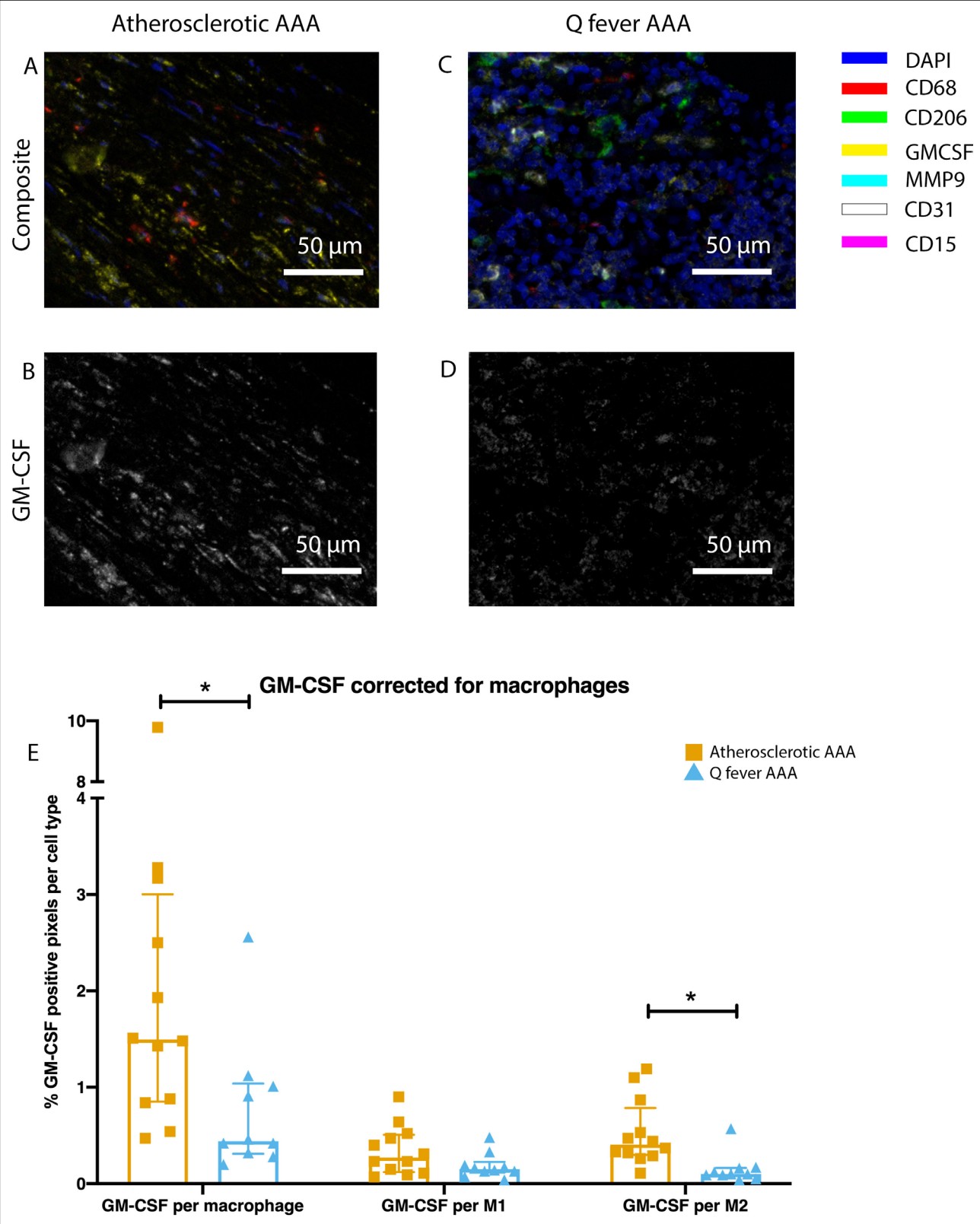

**Figure 6.** Q fever infected AAAs express lower levels of GM-CSF. (**A–D**): Representative composite image of atherosclerotic AAA (**A**) and Q fever AAA (**C**) and corresponding GM-CSF channels (**B, D**). (**E**): The expressed levels of GM-CSF corrected for the number of macrophages and M2 macrophages are lower in Q fever infected AAAs, suggesting an immune-suppressed environment. Source data can be found in *Figure 6—source data 1*.

*Figure 6 continued on next page*

*Figure 6 continued*

The online version of this article includes the following source data for figure 6:

**Source data 1.** Q fever infected AAAs express lower levels of GM-CSF.

area), the numbers of both CD3⁺CD8⁺ cytotoxic T cells and CD3⁺CD8⁻FoxP3⁺ regulatory T cells are increased in Q fever AAA (p=0.043 and p=0.036, respectively) (*Figure 7I*).

## Ruptured and non-ruptured aneurysms exhibit similar immune responses

Strikingly, only Q fever AAAs ruptured while the median diameter (60 mm) did not differ from the diameter in the atherosclerotic AAA group (57 mm) (p=0.887). To investigate whether these patients could be included in our analyses, we tested for differences between ruptured (N=3) and non-ruptured (N=21) aneurysms. Although signs of acute inflammation could be expected, no differences were found in the number of CD3⁺ T cells (p=0.206), CD20⁺ B cells (p=0.407), CD1c⁺ cDC2 (p=0.150), CD15⁺ neutrophils (p=0.275), and CD68⁺ macrophages (p=0.176). Also, when testing for cell subsets, ruptured and non-ruptured aneurysms exhibited similar numbers of CD68⁺CD206⁻ M1 and CD68⁺CD206⁺ M2 macrophages (p=0.206), CD3⁺CD8⁺ cytotoxic T cells (p=0.329), CD3⁺CD8⁻ FoxP3⁺ regulatory T cells (p=0.206), and CD3⁺CD45RO⁺ memory T cells (p=0.176). The similarity of the results between both groups may be explained by the sampling method; all samples were taken from the ventral side of the aneurysm and the rupture side was unknown or not registered at the time.

## Q fever aortas reveal extensive fibrosis

All above-mentioned features are signs of chronic inflammation and long-existing disease. This was supported by HE- and Elastin Van Gieson (EVG) stainings, which demonstrated destruction of elastin fibers and fibrosis. Both atherosclerotic AAAs and Q fever AAAs exhibited extensive atherosclerotic plaque formation. However, there were large differences in vessel architecture as demonstrated in *Figure 8*. Earlier studies have extensively described that aortic aneurysms show fragmentation of elastin fibers indicating media degeneration (*Jana et al., 2019*; *Coady et al., 1999*). In our series, the number of elastin fibers was even more decreased in many Q fever AAAs than in atherosclerotic AAAs with a similar diameter (marked with black arrows in *Figure 8*). In addition, the tunica adventitia showed extensive fibrosis in Q fever AAAs (marked with asterisks in *Figure 8I and L*). These changes indicate the (more pronounced) disrupted architecture in Q fever AAAs, which can attribute to ongoing inflammation.

## Discussion

We are the first to introduce mIHC in vascular manifestations of Q fever to study ongoing local inflammation. This sophisticated mIHC method enabled us to quantify immune cells in large sections of tissue which minimized sampling bias. First, we showed that granulomas are absent in Q fever AAAs. Second, atherosclerotic and Q fever AAAs were similar when comparing the numbers of immune cells. However, there were striking differences in the composition of macrophage- and T cell-phenotypes between AAAs and Q fever AAAs, leading to new insights into the pathogenesis of vascular manifestations of Q fever and its complications and possibly with therapeutic consequences.

Our first observation, the absence of well-formed granuloma formation in our cohort of Q fever AAAs is an important one, since it suggests that the local immune landscape lacks an adequate pro-inflammatory response. In our series, we could not find any well-formed granuloma similar to how they are described in acute Q fever. In acute Q fever manifesting in non-vascular tissue, so-called doughnut granulomas are reported: granulomas with a central clear space and a fibrin ring within or at its periphery (*Maurin and Raoult, 1999*; *Faugaret et al., 2014*), for example, in liver biopsies in case of hepatitis (*Pellegrin et al., 1980*). Here, granuloma is a feature of active defense against the pathogen. However, we should take into account that our group of acute aortitis with Streptococcus species did not show granulomas either. In chronic Q fever granulomas have not been described before (*Faugaret et al., 2014*; *Raoult et al., 2005*). In particular, Lepidi described that resected valve specimens of patients with Q fever endocarditis lacked well-formed granulomas (*Lepidi et al., 2003*).

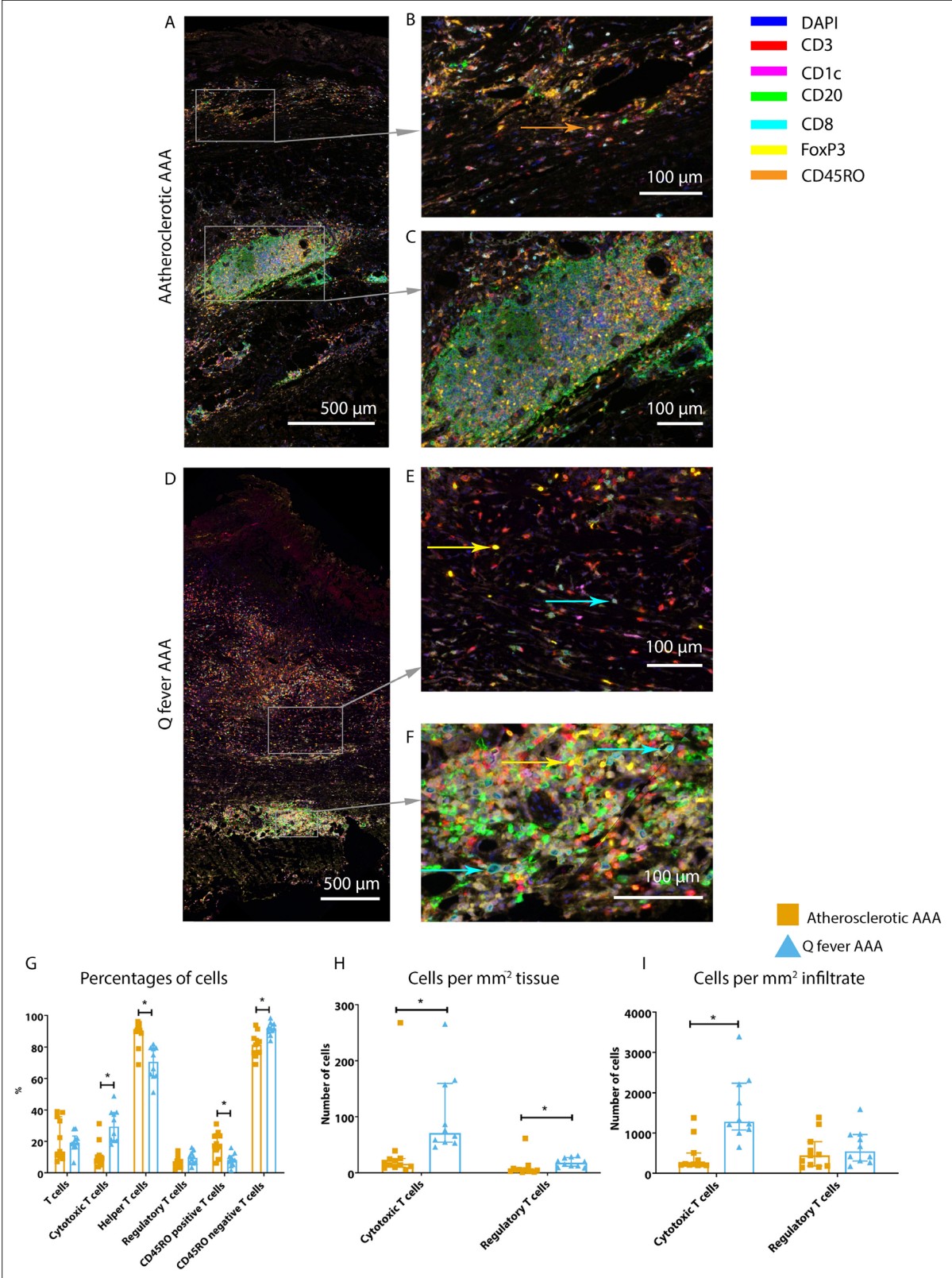

**Figure 7.** Q fever AAAs exhibit both pro-inflammatory and anti-inflammatory T cell subsets. Arrows with corresponding colors indicate the presence of immune cells, with orange for memory T cells, yellow for T helper cells, and cyan for cytotoxic T cells. (**A–F**): Overview of atherosclerotic AAA (**A**) and Q fever AAA with zoomed photos of tissue (**B, E**) and tertiary lymphoid structures (TLS) (**C, F**). In both (**A**) and (**B**), the upper side of the photo is the intima layer. Note all the FoxP3+ (yellow) cells in Q fever infected tissue. (**G**): Percentage of T cells of all cells and T cells subsets out of T cells; (**G, H, I**):

*Figure 7 continued on next page*

*Figure 7 continued*

Quantification shows a shift in cytotoxic/helper T cell ratio and decrease in memory T cells in Q fever AAAs. Q fever AAAs show increased numbers of cytotoxic and regulatory T cells, indicating both immune activation and suppression. Source data can be found in *Figure 7—source data 1*.

The online version of this article includes the following source data for figure 7:

**Source data 1.** Q fever AAAs exhibit both pro-inflammatory and anti-inflammatory T cell subsets.

In vascular Q fever, granulomatous responses consisting of histiocytes surrounding necrotic areas have been reported in Q fever AAAs, however, well-formed granulomas were not found (*Hagenaars et al., 2014a*). Thus, we would interpret the absence of organized granulomas as the first clue for an immune-suppressed environment in AAA of Q fever patients that allows persisting infection after the acute phase.

Second, our results demonstrate some similarities between Q fever AAAs and AAAs. Percentages of CD3[+] T cells, CD20[+] B cells, CD1c[+] cDC2, CD15[+] neutrophils, and CD68[+] macrophages are similar between the groups with atherosclerotic AAA and Q fever AAA. This finding is supported by the PCA, which shows overlapping populations of atherosclerotic and Q fever AAAs when entering these inflammatory cell markers. This does not come as a surprise since there are suggestions that vascular manifestations of Q fever develop in preexisting atherosclerotic aneurysms (*Botelho-Nevers et al., 2007*; *Hagenaars et al., 2014b*; *Hagenaars et al., 2014a*; *Broos et al., 2015*; *Eldin et al., 2017*).

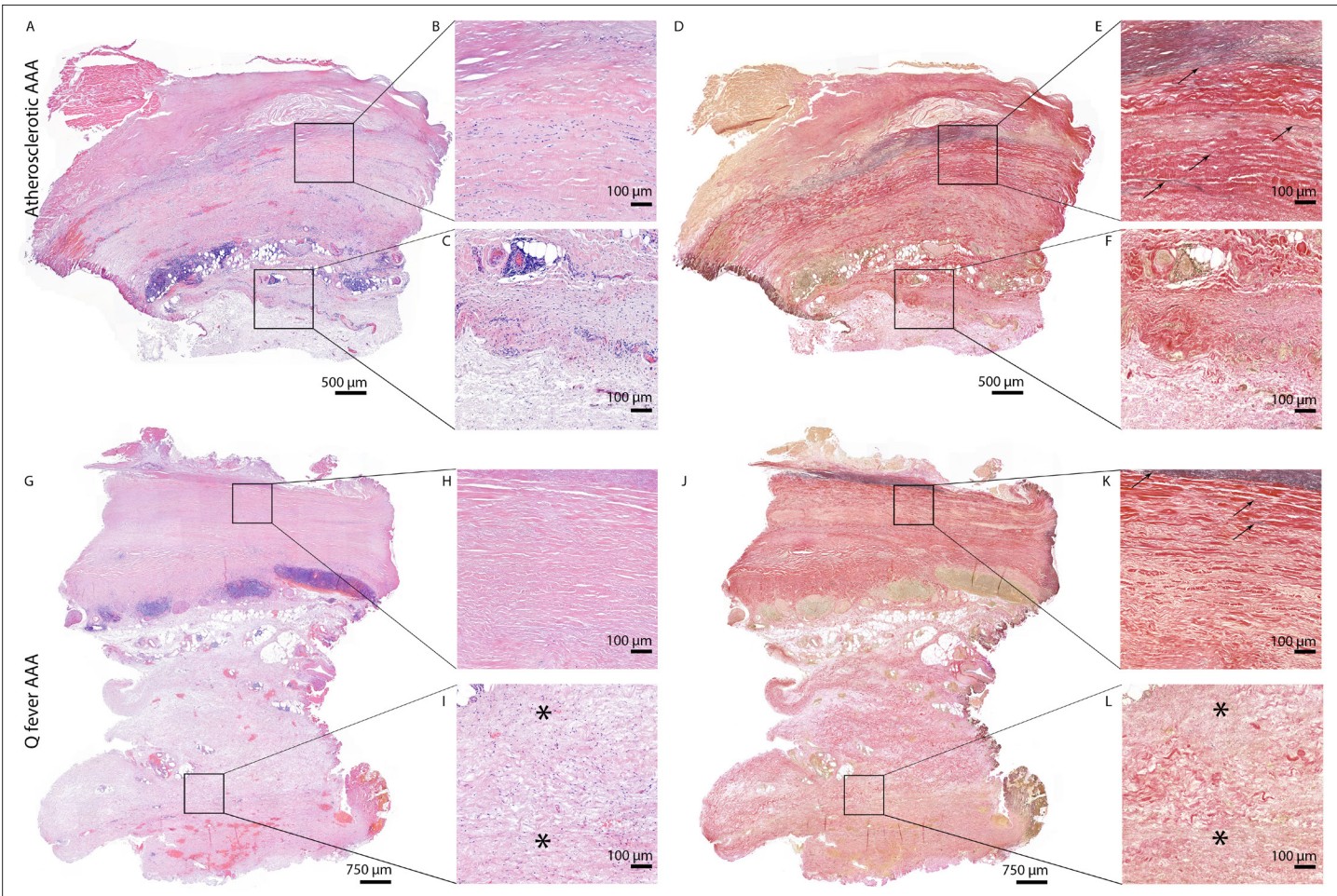

**Figure 8.** HE and Elastin von Gieson (EVG) stainings demonstrate the disrupted architecture of Q fever infected AAAs. Representative images of HE staining of AAA with 22× zoomed-in sections (**A–C**) and EVG staining of adjacent slide (**D–F**) demonstrate the atherosclerotic plaque, immune cells, and infiltrates with relatively preserved vessel architecture as shown by presence of elastin fibers (black arrows pointing at black lines). (HE) (**G–I**) and EVG (**J–L**) of adjacent Q fever AAAs slides reveal pronounced atherosclerosis and immune cell infiltration, and loss of elastin fibers in the media layer (**K**). In the adventitia (**I, L**), tissue is replaced by large amounts of fibrosis, indicated with asterisks.

Despite the similarities, we discovered that atherosclerotic and Q fever AAAs do have important differences, which emerge when investigating macrophage and T cell subset markers. Macrophages in Q fever AAAs were found to be polarized into the less inflammatory M2 phenotype, which is 'tolerogenic' and poorly microbicidal, in contrast to the M1 phenotype that possesses a machinery that can clear an infection. Interestingly, in the AAAs, we found less M2 polarization based on the presence of CD206. This would either indicate that macrophages polarize toward M2 in response to *C. burnetii* infection, or that the presence of M2 polarization is a prerequisite for *C. burnetii* persistence. Previous studies have demonstrated that *C. burnetii* inhabits and proliferates in monocytes and macrophages, and more specifically, in resident vascular wall macrophages in case of vascular Q fever (*Ghigo et al., 2009*; *Lepidi et al., 2003*). It has been shown by *Benoit et al., 2008* that *C. burnetii* stimulates an atypical M2 activation program in monocyte-derived macrophages in vitro (*Benoit et al., 2008*). M2 polarization of macrophages was also observed in *C. burnetii* infected transgenic mice constitutively expressing IL-10 in macrophage lineage, a mouse model for chronic Q fever pathogenesis (*Meghari et al., 2008*). Spleens and livers of these mice showed increased expression of arginase-1 and mannose receptor (CD206) and decreased expression of iNOS, IL-12, and IL-23 in bone marrow-derived macrophages after infection with *C. burnettii* compared to *C. burnetii*-infected wild-type mice. These previous findings suggested that chronic Q fever is associated with M2 polarization of macrophages, but direct evidence in chronic Q fever patients was lacking. Our findings establish that outgrowth and persistence of *C. burnetii* in AAAs is associated with the predominance of CD68+CD206+ M2 macrophages.

There are several possible explanations for the lack of macrophage activation. First, our results demonstrate decreased expression of GM-CSF in Q fever AAAs compared to AAAs. GM-CSF is a pro-inflammatory cytokine that activates granulocytes and macrophages (*Hercus et al., 2012*). Its decreased expression in Q fever AAAs may contribute to the immune-suppressive environment in Q fever AAA. The role of GM-CSF in the context of aneurysm formation has been investigated previously (*Son et al., 2015*). Strikingly, Son et al. described the increased occurrence of aortic dissection and intramural hematoma in wild-type mice subjected to aortic inflammation (CaCl2+ Ang II administration) when also receiving GM-CSF. Only administrating GM-CSF, without the prerequisite of aortic inflammation, did not result in aortic dissection or intramural hematoma. Its potential clinical relevance was confirmed in human blood: GM-CSF serum levels of patients suffering from acute dissection were higher than controls with coronary artery disease, aortic aneurysms or healthy volunteers (*Son et al., 2015*). Additionally, in our cohort, we found that Q fever AAAs ruptured at smaller diameter compared to atherosclerotic AAA. This finding, combined with the GM-CSF paradox, suggests that the development of Q fever AAAs and atherosclerotic AAAs follow different pathways, however strictly hypothetically.

Second, a key cytokine in activation is IFNg, a T-helper (Th)-1 cytokine that activates macrophages and makes them more microbicidal. Previous studies from our group have demonstrated that peripheral blood mononuclear cells from patients with chronic Q fever exhibit an abundant production of IFNg when exposed to *C. burnetii* antigens (*Schoffelen et al., 2014*; *Schoffelen et al., 2017*). These findings were enigmatic since there is an apparent inability of the patient's immune system to kill *C. burnetii* at the infected sites. The current findings would be compatible with a downregulated IFNg response at the infected site.

In addition to differences in macrophage subsets, differences in T cell subsets were also observed. Numerous T cells were observed both in tissue and in TLS, of which the latter are known ectopic lymphoid tissues at inflammation sites, including infections and auto-immune diseases such as atherosclerosis (*Akhavanpoor et al., 2014*; *Akhavanpoor et al., 2018*; *Jing and Choi, 2016*). These structures are important inductive sites for T cells and antibody production (*Jing and Choi, 2016*; *Polverino et al., 2016*; *Humby et al., 2009*). The first difference is the number of CD3+CD8+ cytotoxic T cells, which increased in both infiltrate and surrounding tissue of Q fever AAA compared to AAA. Although the numbers of cytotoxic T cells were high, their function might be compromised, resulting in defective elimination of *C. burnetii*. The increased numbers of CD3+CD8−FoxP3+ regulatory T cells we found in Q fever AAAs may play a role here. An increased number of circulating regulatory T cells has also been shown by *Layez et al., 2012* in Q fever endocarditis patients and in acute Q fever patients (*Layez et al., 2012*). Regulatory T cells can inhibit cytotoxic T cells directly or indirectly (*Joosten and Ottenhoff, 2008*), with a possible role for IL-10 produced by this T cell subset. An important role of

IL-10 in chronic development of Q fever has been postulated based on converging evidence from a series of in vitro studies. IL-10 production by peripheral blood mononuclear cells from patients with Q fever endocarditis and Q fever with valvulopathy who were at risk for developing chronic Q fever was high, compared to control individuals (*Capo et al., 1996*; *Honstettre et al., 2003*). Moreover, IL-10 specifically increases *C. burnetii* replication in naive monocytes (*Ghigo et al., 2001*) possibly by downregulating IFNg. Finally, low IL-10 production in monocytes from patients with acute Q fever was associated with *C. burnetii* elimination, whereas *C. burnetii* replicated in monocytes from patients with chronic Q fever and high IL-10 production. The microbicidal activity of monocytes from patients with chronic Q fever was restored by neutralizing IL-10 (*Ghigo et al., 2004*). The murine model of chronic Q fever mentioned above, also confirmed a key role for IL-10 in bacterial persistence. *C. burnetii* infection is persistent in mice that overexpress IL-10 in the macrophage compartment (*Meghari et al., 2008*). Thus, IL-10 could play a crucial role in this immune-suppressed environment.

The last major difference between Q fever AAA and atherosclerotic AAA is the extent of damage to the vascular wall architecture in Q fever AAAs. This is demonstrated by extensive loss of elastin fibers and increase of fibrosis present in the vascular wall. Fragmentation of elastin fibers has been described for AAAs previously (*Coady et al., 1999*); however, we found the loss of elastin fibers more evident in the lesions from Q fever AAAs than atherosclerotic AAAs. Fibrosis is characterized by replacement of normal tissue by excessive connective tissue and usually follows chronic inflammation. Presence of fibrosis is a sign of a type 2 immune response (*Gieseck et al., 2018*), which we also demonstrated in our cohort with the abundance of M2 macrophages. This may be the effect of persistent presence of growth factors, proteolytic enzymes, angiogenic factors, and profibrotic cytokines (*Wynn, 2008*; *Wynn, 2007*). Previously, fibrosis was also observed in chronic Q fever endocarditis in humans and cows (*Lepidi et al., 2003*; *Agerholm et al., 2017*; *Brouqui et al., 1994*; *De Biase et al., 2018*). This indicates that our Q fever AAA cohort suffered from more destructive disease than our AAA cohort.

These novel insights could lead to new clues for novel treatments and thus developments for clinical care. Currently, Q fever AAA still leads to significant morbidity and mortality rates despite antibiotic and surgical treatment. Epidemiological studies demonstrate the similar risk profile of Q fever and non-Q fever infected AAAs, yet the risk of complications is higher in the Q fever infected group (*Hagenaars et al., 2014b*), even up to 61% (*van Roeden et al., 2019*), and 25% of patients suffering from Q fever AAA had deceased with a definitely/probably chronic Q fever related cause of death (*van Roeden et al., 2019*). Here, we confirm that in vascular manifestations of Q fever, the local immune response is skewed toward an immunotolerant state. Hypothetically, the decreased expression of GM-CSF suggests a possible role for immunomodulating treatment, for example, with administration of recombinant GM-CSF. This is already approved for neutropenia due to myelosuppression (*Dougan et al., 2019*), and has been suggested for treatment for pulmonary tuberculosis (*Damiani et al., 2020*). There might be a role for immunomodulating adjuvant therapies in patients with Q fever AAA in whom treatment failure is observed with antibiotics alone.

Our study was the first to use mIHC in Q fever AAA and thereby to gain information about the number and proportion of immune cells, and simultaneously obtain spatial information. This powerful technique and the access to rare Q fever AAA tissue are strengths of this study. While other studies have tested for immune cell activation and recruitment in peripheral blood, we were able to study the actual infected tissue. Interpreting our results in context of previous observations enables us to increase our understanding of the pathophysiology of Q fever AAA. Still, several limitations should be noted. First, our sample size is limited with only 10 vascular manifestations of Q fever samples. However, this is still the largest study investigating local immune responses in Q fever AAA in humans. In addition, in our quantification method, we include entire slides up to 238 20× views per patient, which minimizes the effects of the small sample size. Second, consistent with IHC studies in general, we can only describe the immune cells we observe, without answering mechanistic questions. Although epidemiological studies suggest that Q fever AAA is the result of *C. burnetii* infected AAAs, our study did not support causation. Moreover, our study lacks direct IHC identification of *C. burnetii* in Q fever AAA, however, all samples were proven PCR positive. Additionally, elastin (breakdown) and fibrosis were not quantified. Finally, it should be emphasized that these results and interpretations are based on tissue samples from patients with indication for surgery. Nevertheless, when interpreting our results in light of the current literature, we can reasonably formulate hypotheses about the pathophysiology and test these in further research.

Taken together, this leads to the following hypothesis with a prominent role for immune suppression. First, macrophages that harbor *C. burnetii* are not effectively killing the micro-organisms, probably due to a lack of activation by proinflammatory cytokines like GM-CSF and IFN-g in a microenvironment with excess IL-10. Second, effector T cells that attempt to eliminate the intracellular bacterium residing in monocytes and macrophages, are hindered by regulatory T cells that are prominent IL-10 producers. Third, there is a lack of microbicidal M1 macrophages, instead macrophages are polarized into the tolerogenic M2 phenotype, which leads to insufficient attack of the pathogen, enabling persistent infection.

## Acknowledgements

The authors would like to acknowledge Anne van Duffelen and Kiek Verrijp for their help in optimizing the staining procedure, Jelena Meek and Ine van Raaij for assistance in staining, Jan Damen for conventional stains and archive management, Inge Wortel, Shabaz Sultan, and Johannes Textor for their help in data analysis, Janneke Timmermans for her feedback, and Sandra Vreman for discussion. This work was supported by SCAN consortium: European Research Area - CardioVascualar Diseases (ERA-CVD) Grant [JTC2017-044] and TTW-NWO open technology Grant [STW-14716].

## Additional information

### Competing interests

Jos W Van der Meer: Senior editor, *eLife*. The other authors declare that no competing interests exist.

### Funding

| Funder | Grant reference number | Author |
| --- | --- | --- |
| European Research Area Network on Cardiovascular Diseases | JTC2017-044 | Kimberley RG Cortenbach |
| TTW-NWO Open Technology | STW-14716 | Alexander HJ Staal |

The funders had no role in study design, data collection and interpretation, or the decision to submit the work for publication.

### Author contributions

Kimberley RG Cortenbach, Conceptualization, Formal analysis, Visualization, Writing – original draft; Alexander HJ Staal, Conceptualization, Formal analysis, Investigation, Writing – review and editing; Teske Schoffelen, Anne FM Jansen, Chantal P Bleeker-Rovers, Conceptualization, Methodology, Writing – review and editing; Mark AJ Gorris, Lieke L Van der Woude, Investigation, Methodology, Writing – review and editing; Paul Poyck, Robert Jan Van Suylen, Conceptualization, Resources, Writing – review and editing; Peter C Wever, Conceptualization, Investigation, Resources; Mangala Srinivas, Funding acquisition, Supervision, Writing – review and editing; Konnie M Hebeda, Investigation, Supervision, Writing – review and editing; Marcel van Deuren, Conceptualization, Writing – review and editing; Jos W Van der Meer, Conceptualization, Supervision, Writing – review and editing; Jolanda M De Vries, Supervision, Writing – review and editing; Roland RJ Van Kimmenade, Conceptualization, Methodology, Supervision, Writing – review and editing

### Author ORCIDs

Kimberley RG Cortenbach http://orcid.org/0000-0002-2717-5527
Konnie M Hebeda http://orcid.org/0000-0002-4181-3302
Roland RJ Van Kimmenade http://orcid.org/0000-0002-8207-8906

### Ethics

Human subjects: The medical ethics committees of the institutions approved the study, in line with the principles outlined in the Declaration of Helsinki (Radboudumc: 2017-3196; Jeroen Bosch Hospital: 2019.05.02.01).

Decision letter and Author response
Decision letter https://doi.org/10.7554/eLife.72486.sa1
Author response https://doi.org/10.7554/eLife.72486.sa2

## Additional files

### Supplementary files
• Supplementary file 1. Loadings of Principal Component Analysis and Overview of Reagents and dilutions.

• Transparent reporting form

### Data availability
All data generated or analyzed during this study are included in the manuscript and uploaded to Dryad (http://dx.doi.org/10.5061/dryad.bzkh189b4). Figure 3 - Source data 3; Figure 5 - Source data 5; Figure 6 - Source figure 6; Figure 7 - Source figure 7 contain numerical data used to generate the figures.

The following dataset was generated:

| Author(s) | Year | Dataset title | Dataset URL | Database and Identifier |
|---|---|---|---|---|
| Cortenbach KR | 2021 | Vascular Q fever inflammation | http://dx.doi.org/10.5061/dryad.bzkh189b4 | Dryad Digital Repository, 10.5061/dryad.bzkh189b4 |

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
