## [Editor Report]

This is a collaborative study of clinical centers that investigates tissue pathology and immune cell infiltration of aortic aneurysms from chronic Q fever patients. The combination of precious and rare human tissue samples with well-designed multiplex IHC panels for characterizing local immune responses within the spatial context is unique in the field of human infection immunology and has revealed unprecedented insight into the manifestation of this disease.

---

## [Decision Letter]

**Decision letter after peer review:**

Thank you for submitting your article "Multiplex immunohistochemistry differences between Q fever and atherosclerotic abdominal aortic aneurysms indicate immune suppression" for consideration by *eLife*. Your article has been reviewed by 2 peer reviewers, and the evaluation has been overseen by a Reviewing Editor and a Senior Editor. The following individuals involved in review of your submission have agreed to reveal their identity: Roland Lang (Reviewer #1); Jean-Louis Mege (Reviewer #2).

Essential revisions:

1) Although the authors worked with two 6-marker panels, the extent of multiplexing in the IHC methodology is still limited. Thus, the combination of antibodies used to define immune cell types and subsets or activation states has a limited resolution and leaves out populations of interest. E.g. the differentiation between M2 and M1 macrophages is exclusively made based on CD206 staining; no staining for CD4 T cells (not to speak of Th subsets) was included. Similarly, only GM-CSF was the only cytokine measured. The authors should be careful about defining these populations on the limited markers, for example, macrophages polarization can not only be based on differential expression of CD206. Can other markers be added/used? Otherwise, conclusions and interpretations need to be tempered accordingly.

2) A blind spot in the authors' visual field of the Q fever aneurysm is spatial and quantitative information about the pathogen. While all samples were classified by PCR for C. burnetii as Q fever or control atherosclerotic aneurysms, no IHC of C. burnetii is included in the mIHC panels or in a separate analysis. Thus, information about the types and activation status of C. burnetii-harboring host cells is absent: are the M2-like CD206+ macrophages infected more heavily? Is there a spatial association of immunoregulatory cells (Foxp3+ Treg) with infected host cells? Are C. burnetii also infecting endothelial cells or neutrophils? Is it possible to add staining for C. burnetii in the aneurysms?

3) An inherent limitation of the descriptive nature of the study is that it cannot determine a causal role for specific findings in the development of vascular Q fever. E.g., the higher abundance of M2-like macrophages may be a cause or a consequence of chronic C. burnetii infection. The finding of reduced GM-CSF and increased CD8^+^ T cells in Q fever aneurysms is novel and interesting, but the functional relevance thereof remains unclear. Therefore, it is better to avoid causal conclusions like the last sentence of the abstract.

4) The lack of quantification for elastin destruction and fibrosis (Figure 6) is surprising since it does not keep up with spirit of the rest of the manuscript. The authors should provide also in this Figure a quantitative assessment of fibrosis and elastin breakdown. Development of fibrosis is a hallmark of type 2 immune bias (e.g. liver fibrosis in schistosoma infection). Given the abundance of M2-like macrophages in the aortic tissue in vascular Q fever, I wonder whether this potential link to elastin breakdown and fibrosis should not be made in the Discussion section.

5) The authors did not observe granuloma structures in the Q fever aneurysms. This finding is consistent with earlier reports on the histopathology of heart valves in chronic Q fever. It is stressed in the manuscript that the lack of granulomas is indicative of the immunosuppressive environment. However, the comparison with the hepatic, dough-nut granulomas in acute infection is problematic because of the completely different tissue types. The correct comparison would in fact be to use aortic tissue from another infection and to determine whether granulomas are formed in the vessel wall. Was there formation of granulomas in the aneurysms from acute bacterial infection?

6) Is it clear that all Q fever aneurysms are the result of super-infection of existing atherosclerotic aneurysms with C. burnetii, as presented in the Discussion? Would this statement not require the demonstration of the aneurysms prior to C. burnetii infection? How strong is the evidence to exclude that aneurysm formation is a consequence of C. burnetii infection?

7) The title provides a partially correct conclusion of the results. If there is a signature in favor of M2 polarization in Q fever AAA, this is not the demonstration of immune suppression. The authors should re-write the title to better represent the findings of the manuscript. The authors should also temper similar claims throughout the manuscript.

8) The authors should describe how they calculated the number of patients necessary to conduct the study. In addition, the proportion of men in infectious AAA is only 50% whereas it was 83% in atherosclerotic AAA. Is this a consequence of the small size of infectious AAA sample. What is specific to infectious AAA as compared with atherosclerosis AAA?

9) The authors should quantify tertiary lymphoid structures and describe why they consider them as a measure of lymphocyte proliferation.

The small size of infectious AAA sample is a source of conclusion. It would be better to include these results in additional material.

[Editors’ note: further revisions were suggested prior to acceptance, as described below.]

Thank you for resubmitting your work entitled "Differences in local immune cell landscape between Q fever and atherosclerotic abdominal aortic aneurysms identified by multiplex immunohistochemistry" for further consideration by *eLife*. Your revised article has been evaluated by a Reviewing Editor and a Senior Editor.

The manuscript has been improved but there are some remaining issues that need to be addressed, as outlined below:

1. Line 64: " suffer from renal insufficiency or aneurysm," This sentence is factually incorrect since the kidney does not develop aneurysms – only blood vessels do. The authors need to indicate what they mean (likely aortic aneurysm) and also provide a corresponding reference.

2. Line 69: The sentence "Vascular Q fever has severe clinical consequences" is not quite accurate, since not all patients are affected equally. It should be rephrased to "Vascular manifestations of Q fever can have severe clinical consequences". Also, since "vascular Q fever" also includes heart valves (which was not studied here) it should also be replaced with "vascular manifestations of Q fever" in lines 66, 83, 93, 97, 104, 264, 302, 307, 331, 342, and 430.

3. Line 71: "acute aneurysms". There is no such thing as an "acute aneurysm". However, patients with aortic aneurysms may experience acute complications, such as rupture which carries a high mortality. This needs rewording.

4. Line 93 ff: The authors do not indicate that the aneurysm tissue studied actually were obtained from human patients, which greatly increases the clinical importance and implications of this work.

5. Line 101 ff: There is no information about the "control" group patients, how the tissues were obtained, whether they had aortic atherosclerosis, aneurysms or not etc. A description of gross pathology of the specimens needs to be included for each patient studied, e.g. in a table. Also, for all groups the age of the patients included needs to be provided. This information is essential and needs to be added.

6. Line 289 "Q fever aortas reveal extensive fibrosis" This is a remarkable and novel finding which at this point cannot be explained by the locally suppressed micro-environment observed by the author, and actually would be counterintuitive. The authors should discuss this rather paradoxical finding with regard to the potential mechanisms involved.

---

## [Author Response]

Essential revisions:1) Although the authors worked with two 6-marker panels, the extent of multiplexing in the IHC methodology is still limited. Thus, the combination of antibodies used to define immune cell types and subsets or activation states has a limited resolution and leaves out populations of interest. E.g. the differentiation between M2 and M1 macrophages is exclusively made based on CD206 staining; no staining for CD4 T cells (not to speak of Th subsets) was included. Similarly, only GM-CSF was the only cytokine measured. The authors should be careful about defining these populations on the limited markers, for example, macrophages polarization can not only be based on differential expression of CD206. Can other markers be added/used? Otherwise, conclusions and interpretations need to be tempered accordingly.

We totally agree with the reviewers regarding cell (subset) populations. When we designed these panels, we decided to create panels for studying the overview of immune cells. More in-depth characterization of macrophages and T helper cells and their cytokines would be an excellent follow-up study.

We are aware of the limitations of the chosen markers in this manuscript. We acknowledge that we cannot distinguish separate helper T cell populations and that CD206 is not the only marker for macrophage polarization. Therefore, we made substantial revisions to our manuscript in the results and Discussion sections.

Page/line number: page 7 line 17; page 8 line 5-24; page 11 line 12-13; page 12 line 2; page 13 line 69.

2) A blind spot in the authors' visual field of the Q fever aneurysm is spatial and quantitative information about the pathogen. While all samples were classified by PCR for C. burnetii as Q fever or control atherosclerotic aneurysms, no IHC of C. burnetii is included in the mIHC panels or in a separate analysis. Thus, information about the types and activation status of C. burnetii-harboring host cells is absent: are the M2-like CD206+ macrophages infected more heavily? Is there a spatial association of immunoregulatory cells (Foxp3+ Treg) with infected host cells? Are C. burnetii also infecting endothelial cells or neutrophils? Is it possible to add staining for C. burnetii in the aneurysms?

We thank the reviewer for this valuable comment. We agree that spatial information about the pathogen would be a valuable addition to this manuscript. We would like to emphasize that the samples are PCR positive, which is a very sensitive method for proving the presence of C. burnetii. Literature applying IHC for C. burnetii is scarce, and after discussion with experts, also from other centers, we concluded specific antibodies were available (1), but unfortunately, not anymore. In literature some stainings have been used, amongst others the Kinyoun and Auramine method (2, 3). Both are methods to demonstrate acid-fast bacteria. Kinyoun is a modified Ziehl-Neelsen staining and visible in brightfield, Auramine-Rhodamine is a fluorescent staining visible in the Texas Red channel.

In optimization of these experiments, we noticed that using the Kinyoun method, identification of the bacteria was very difficult because of their small size and poor contrast to the surrounding tissue. Therefore we decided to stain our tissue using the Auramine method, since fluorescence provides better contrast, which would help us in finding the bacteria in the Kinyoun-stained tissue. During optimization of the Auramine experiments, we noticed that the IHC and DAPI were influenced by the auramine staining, and order experiments have pointed out that the best results were gained by (1) IHC followed by (2) DAPI and scanning, and then the same sample underwent (3) Auramine staining and repeatedly (4) DAPI followed by the final scan. Additionally we only applied Auramine staining on a consecutive slide.

Author response image 1 shows the positive control, which is a lymph node from a goat with an acid-fast bacterium inhabiting. It is clear that Auramine is in high numbers present in both methods of staining.Author response image 2 shows a magnification of an intima layer of the aorta which shows only one Auramine-positive spot. This spot is not colocalizing with CD68. Additionally, only treating the sample with Auramine-staining on a consecutive slide did not reveal more positive areas. When we use this information to study the Kinyoun staining, we did not find any bacterium-like structures. Author response image 3, shows a magnification of an adventitia layer of the aorta. Here are some more Auramine-positive spots but again, these are scarce compared to our positive control. In the Kinyoun staining we could not find bacterium-like structures.

**Author response image 1. sa2fig1:** Positive control for Auramin staining. Tissue was stained with both IHC and Auramine (first row), only with Auramine (middle row), or scanned between the IHC and Auramine steps (last row). Auramine channel shows numerous positive areas in the infected tissue (goat lymph node). All scale bars represent 50 µm.

**Author response image 2. sa2fig2:** Auramine and Kinyoun staining on Q fever infected AAA, magnification in adventitia. Tissue was stained with both IHC and Auramine (first row), only with Auramine (middle row), or scanned between the IHC and Auramine steps (last row). Some Auramine-positive spots are present (red arrows). In the IHC only and Kinyoun images these arrows point at corresponding locations. In Kinyoun staining no bacterium was found. All scale bars represent 50 µm..

**Author response image 3. sa2fig3:** Auramine and Kinyoun staining on Q fever infected AAA, magnification in intima. Tissue was stained with both IHC and Auramine (first row), only with Auramine (middle row), or scanned between the IHC and Auramine steps (last row). Some Auramine-positive spots are present (red arrows). In the IHC only and Kinyoun images these arrows point at corresponding locations. In Kinyoun staining no bacterium was found. All scale bars represent 50 µm..

As a negative control, we included one or our normal aortas. Unfortunately, in this tissue we also found a Auramine-positive structure (Author response image 4), with the same shape as shown before.

**Author response image 4. sa2fig4:** Auramine staining on normal aorta, magnification in media. In media an Auramine-positive structure was found, in granular shape similar to described in earlier figures. All scale bars represent 50 μm.

Concluding: we attempted to identify *C. burnetii* implementing staining techniques as described in literature but we were not able to obtain reliable results, as also mentioned in literature. We believe that PCR is the best option at this point of time and we mention the lack of direct staining in the limitations. We think that these results might give directions for future research but currently we are very cautious drawing conclusions. Additionally, it is known that IHC of pathogens involved in chronic infections can be very challenging, as demonstrated earlier in tuberculosis. We have added the limitation of lacking of *C. burnetii* direct staining in the limitations. Page/line number: page 15 line 13-14.

3) An inherent limitation of the descriptive nature of the study is that it cannot determine a causal role for specific findings in the development of vascular Q fever. E.g., the higher abundance of M2-like macrophages may be a cause or a consequence of chronic C. burnetii infection. The finding of reduced GM-CSF and increased CD8^+^ T cells in Q fever aneurysms is novel and interesting, but the functional relevance thereof remains unclear. Therefore, it is better to avoid causal conclusions like the last sentence of the abstract.

We thank the reviewer for this comment. We acknowledge that we cannot draw causal conclusions about disease progression and development. Therefore, we have made changes in our abstract and discussion to avoid these causal statements. Page/line number: page 3 line 15-16; page 16 line 12-14.

4) The lack of quantification for elastin destruction and fibrosis (Figure 6) is surprising since it does not keep up with spirit of the rest of the manuscript. The authors should provide also in this Figure a quantitative assessment of fibrosis and elastin breakdown. Development of fibrosis is a hallmark of type 2 immune bias (e.g. liver fibrosis in schistosoma infection). Given the abundance of M2-like macrophages in the aortic tissue in vascular Q fever, I wonder whether this potential link to elastin breakdown and fibrosis should not be made in the Discussion section.

We want to thank the reviewer for this comment. We agree that quantification is the best method of presenting our data. However there were difficulties in quantification of fibrosis, which hampered us from this analysis. We have presented an overview of this issue in Author response image 5 in this response letter below. Even at high level magnification and use of both HE and EVG staining, it was not possible to accurately draw borders between fibrosis and non-fibrotic tissue. In addition, we considered additional stainings to address this question. However, fibrotic tissue and healthy adventitia both consist of collagen and thus a fibrosis staining, showing the presence of collagen, is not able to distinguish between pre-existent and fibrotic adventitia and will not solve the problem of the difficulty of drawing borders between fibrotic and non-fibrotic tissue. Nevertheless, although we were not able to quantify this, this finding was too interesting not to describe in this manuscript and is still supportive to the main message, and hopefully this could inspire future work.

**Author response image 5. sa2fig5:** HE (A) and EVG (B) staining of Q fever AAA demonstrate damaged vessel wall architecture. EVG stained images (B) demonstrate fine fibrotic areas (marked with *) enclosed by pre-existent adventitial structures (marked with ^), mainly consisting of collagen bundles and smooth muscle cells.

We would like thank the reviewer for the excellent comment regarding the type 2 immune bias in relation to M2 macrophages. We have added this valuable argument to the discussion. Page/line number: page 14 line 6-8.

Secondly, we also would like to address the second part of the reviewer’s comment, which is the quantification of elastin fibers and/or destruction. We completely agree that quantification of this described trend would be a valuable addition for our manuscript. Therefore we attempted to quantify the amount of elastin fibers in our samples based on the EVG stained images. To do so, we have tried different methods as shown in Author response image 6 and Author response image 7.

**Author response image 6. sa2fig6:** Elastin analysis in inForm software. A-C: Representative images of training regions (drawn colored regions in green, blue, and red), which are very small in B and therefore indicated with arrows with corresponding colors. The segmented images show nice segmentation of background, tissue and elastin. D-F: These images demonstrate the imperfections of the algorithm: not all elastin fibers are recognized as indicated with blue arrows.

**Author response image 7. sa2fig7:** Elastin analysis in FIJI. The original stained image (A) was converted into RGB (B). C-H: Color deconvolution was performed, which resulted in these segmented images with corresponding thresholds. For all options it was not possible to exclude cells in this threshold while including elastin, resulting in an unreliable analysis.

Author response image 6 demonstrates the analysis in inForm, which is the software in which we have performed all other analyses. inForm allows so-called tissue segmentation, in which regions are drawn for training, and after training the algorithm will segment all images based on this specific training. Image 4ABC demonstrate some representative training regions and the segmentation after training on these specific images. In these examples the different regions (background, tissue, and elastin) are recognized well. However, Author response image 6 is more representative for the majority of the images. Not all elastin fibers are recognized well which results in underestimation of the presence of these fibers.

Therefore we decided to continue our analysis in FIJI. Our protocol is based on literature describing brightfield threshold analysis. (4) First, the image of interest Author response image 7 was converted into RGB format Author response image 7. To be able to perform threshold analysis, color deconvolution needed to take place for which this plugin was downloaded (https://blog.bham.ac.uk/intellimic/g-landinisoftware/colour-deconvolution-2/). However, EVG staining was not an option in the color deconvolution menu, thus we have selected several ‘second best’ options Author response image 7. These deconvoluted images contained the highest contrast between elastin and non-elastin. After applying the Threshold Tool on all of these images, we noted that in all samples there was no optimal threshold which included (almost) all elastin without excluding (almost) all cells in the infiltrate.

In conclusion: both the inForm and FIJI method did not provide the reliable elastin quantification we were aiming for. Therefore, we decided not to include any of these analyses in our manuscript but we still believe that this trend is an interesting addition to our manuscript. We made changes to the limitations section to acknowledge the lack of the quantification of fibrosis and elastin. Page/line number: page 15 line 14-15

5) The authors did not observe granuloma structures in the Q fever aneurysms. This finding is consistent with earlier reports on the histopathology of heart valves in chronic Q fever. It is stressed in the manuscript that the lack of granulomas is indicative of the immunosuppressive environment. However, the comparison with the hepatic, dough-nut granulomas in acute infection is problematic because of the completely different tissue types. The correct comparison would in fact be to use aortic tissue from another infection and to determine whether granulomas are formed in the vessel wall. Was there formation of granulomas in the aneurysms from acute bacterial infection?

Thank you for your comment. Indeed, the comparison with acutely infected aortic tissue would be better than the non-related tissue. We have implemented this suggestion in the Discussion section. Page/line number: page 10 line 13-17.

6) Is it clear that all Q fever aneurysms are the result of super-infection of existing atherosclerotic aneurysms with C. burnetii, as presented in the Discussion? Would this statement not require the demonstration of the aneurysms prior to C. burnetii infection? How strong is the evidence to exclude that aneurysm formation is a consequence of C. burnetii infection?

The reviewer remarks an excellent point. We do think that epidemiological studies can support this statement, however, we agree that there not conclusive evidence supporting this statement. We made adjustments to the Discussion section accordingly.

Page/line number: page 12 line 5, page 15 line 12-16.

7) The title provides a partially correct conclusion of the results. If there is a signature in favor of M2 polarization in Q fever AAA, this is not the demonstration of immune suppression. The authors should re-write the title to better represent the findings of the manuscript. The authors should also temper similar claims throughout the manuscript.

We would like to thank the reviewer for this comment. Throughout the manuscript and in the title, we have tempered claims about immune suppression. Page/line number: page 1 line 1-2, page 3 line 15-16, page 11 line 12-13, page 12 line 2, page 15 line 12-16.

8) The authors should describe how they calculated the number of patients necessary to conduct the study. In addition, the proportion of men in infectious AAA is only 50% whereas it was 83% in atherosclerotic AAA. Is this a consequence of the small size of infectious AAA sample. What is specific to infectious AAA as compared with atherosclerosis AAA?

We would like to thank the reviewer to point out this important issue. We agree that proper sample size calculation should be performed in order to execute robust statistical analysis.

In our study, we included rare tissue samples. Therefore we included all available samples for these rare groups: normal abdominal aortas, Q fever AAA and acute infection of AAA.

To determine how many atherosclerotic AAAS should be included, we performed a sample size calculation based on preliminary data. This calculation is added to the ‘Transparent Reporting Form’.

Moreover, the difference between infectious AAA and atherosclerotic AAA is the PCR proven presence of pathogen in the infectious AAA.

We are aware that there may be a bias in the included samples: only subjects are included with an indication for surgery, this applies to all four groups. We have added the sample size calculation to the and a description of this potential bias to the limitations in the Discussion section. Page/line number: page 15 line 15-16.

9) The authors should quantify tertiary lymphoid structures and describe why they consider them as a measure of lymphocyte proliferation.The small size of infectious AAA sample is a source of conclusion. It would be better to include these results in additional material.

We would like to thank the reviewer for this comment. We agree that TLS should be quantified, so we have added this information to the Results section. In addition we have added additional information about TLS in the Discussion section and the definitions for TLS in the method section. We agree that the sample size of infectious AAA is very limited. We clarified immune cell infiltration in infectious AAA in figure 1, so we believe that further clarification in the supplemental section is not needed anymore.

References

1. Roest H-J, van Gelderen B, Dinkla A, Frangoulidis D, van Zijderveld F, Rebel J, et al. Q fever in pregnant goats: pathogenesis and excretion of Coxiella burnetii. PloS one. 2012;7(11):e48949-e.

2. Leyk W, Scheffler R, Schliesser T. [Direct demonstration of Coxiella burnetii from air samples on slides coated with kraton 1107]. Zentralbl Bakteriol Orig A. 1976;234(1):105-9.

3. Moore JD, Barr BC, Daft BM, O'Connor MT. Pathology and diagnosis of Coxiella burnetii infection in a goat herd. Vet Pathol. 1991;28(1):81-4.

4. Surman TL, Abrahams JM, Manavis J, Finnie J, O’Rourke D, Reynolds KJ, et al. Histological regional analysis of the aortic root and thoracic ascending aorta: a complete analysis of aneurysms from root to arch. Journal of Cardiothoracic Surgery. 2021;16(1):255.

[Editors’ note: further revisions were suggested prior to acceptance, as described below.]

The manuscript has been improved but there are some remaining issues that need to be addressed, as outlined below:1. Line 64: " suffer from renal insufficiency or aneurysm," This sentence is factually incorrect since the kidney does not develop aneurysms – only blood vessels do. The authors need to indicate what they mean (likely aortic aneurysm) and also provide a corresponding reference.

We like to thank the reviewer for this sharp observation. We have changed the sentence accordingly and we have added a reference as follows:

“Progression to chronic Q fever occurs in approximately 5% of infected individuals (2); people at risk are older, suffer from aortic or iliac aneurysm or renal insufficiency (3), or previously underwent valvular or vascular prosthesis surgery. (3)”

2. Line 69: The sentence "Vascular Q fever has severe clinical consequences" is not quite accurate, since not all patients are affected equally. It should be rephrased to "Vascular manifestations of Q fever can have severe clinical consequences". Also, since "vascular Q fever" also includes heart valves (which was not studied here) it should also be replaced with "vascular manifestations of Q fever" in lines 66, 83, 93, 97, 104, 264, 302, 307, 331, 342, and 430.

Thank you for this comment, we agree that adjustment of this sentence and term improves the manuscript. We changed this sentence into ‘Vascular manifestations of Q fever can have severe clinical consequences’ and replaced ‘vascular Q fever’ by ‘vascular manifestations of Q fever’ throughout the manuscript. Line number 66, 67, 81, 91, 96, 103, 263, 264, 301, 307, 330, 417, 430.

3. Line 71: "acute aneurysms". There is no such thing as an "acute aneurysm". However, patients with aortic aneurysms may experience acute complications, such as rupture which carries a high mortality. This needs rewording.

We would like to thank the reviewer for this comment. We have reworded this sentence into:

“Of these, acute complications (i.e. rupture, dissection, endoleak or symptomatic aneurysms) were most prevalent (35%), followed by abscesses (22%), and fistula (14%).” Line number 69.

4. Line 93 ff: The authors do not indicate that the aneurysm tissue studied actually were obtained from human patients, which greatly increases the clinical importance and implications of this work.

Thank you for this comment, this is indeed an important addition which we have added to the study description as follows:

“To address this, we have investigated the local immune response in *C. burnetii*-infected AAAs (Q fever AAA), classical atherosclerotic abdominal aortic aneurysms (AAA), acutely infected AAA, and control aorta tissue, applying multiplex immunohistochemistry (mIHC) on human patient tissues. We investigated both the adaptive and innate immune system.” Line number 94-95.

5. Line 101 ff: There is no information about the "control" group patients, how the tissues were obtained, whether they had aortic atherosclerosis, aneurysms or not etc. A description of gross pathology of the specimens needs to be included for each patient studied, e.g. in a table. Also, for all groups the age of the patients included needs to be provided. This information is essential and needs to be added.

Thank you for pointing out this important issue. To further clarify these groups, we have added a small introduction in line 102, which now introduces the patient groups as following:

“Abdominal aorta tissue samples from patients with Q fever infected aneurysms and control groups (i.e., atherosclerotic AAAs, acutely infected AAAs, and non-aneurysmatic aortas) were investigated with a novel mIHC method to study the involvement of the innate and adaptive immune system in vascular manifestations of Q fever.”

The paragraph ‘Patient Samples’ (line 108-122) describes the groups in more detail and we have added some information about the obtainment of these samples. The first three groups (Q fever AAA, atherosclerotic AAA, and acutely infected AAA) are all aneurysmatic aortic tissue samples, contrary to the normal non-aneurysmatic aortas. Our atherosclerotic AAAs were selected at random from our database. Tissue of acutely infected AAAs and non-aneurysmatic aortas is rather scarce and because of their limited availability, we have included all available and eligible samples for our study.

This information has now been added to this paragraph, resulting in the following:

“Patient samples

Tissue samples were collected from four groups of patients in two Dutch hospitals: Jeroen Bosch Hospitals in ‘s Hertogenbosch and Radboud university medical center in Nijmegen. The first group consisted of patients diagnosed with C. burnetii infected AAA (Q fever AAA) according to the Dutch consensus guideline (15): all patients had an abdominal aneurysm (AAA) and IgG phase I was at least 1:1024 in combination with a positive PCR of aortic tissue. The second group consisted of patients with atherosclerotic AAA without clinical suspicion of Q fever which were selected from out database at random. The third group consisted of patients with an acutely infected AAA, with the same definition of AAA in combination with positive cultures of Streptococcus pneumoniae and Streptococcus Agalactiae, respectively. In these three groups AAA was defined as a CT proven abdominal aortic aneurysm with a diameter of at least 3.0 cm. (16) All aneurysmatic tissue samples were either obtained from patients undergoing elective surgical repair or emergency repair in case of aortic rupture. The fourth group consisted of abdominal aorta samples from patients undergoing kidney explantation surgery for transplantation purposes, with an aortic diameter smaller than 3.0 cm. Due to the limited availability of acutely infected AAAs and non-atherosclerotic aortas, all available and eligible samples were selected. The samples from Jeroen Bosch Hospital were described in a previous study. (17)”.

In addition, Table 1 contains descriptive information about all included groups (page 36). Here, we elaborate on sex, age, diameter on CT and various clinical baseline characteristics of all groups. As requested, we have added all ages to this table.

We hope that this information answers your question. Please let us know if we did not interpret your question correctly. By all means, we are willing to further improve our manuscript.

6. Line 289 "Q fever aortas reveal extensive fibrosis" This is a remarkable and novel finding which at this point cannot be explained by the locally suppressed micro-environment observed by the author, and actually would be counterintuitive. The authors should discuss this rather paradoxical finding with regard to the potential mechanisms involved.

We would like to thank the reviewer for this valuable comment. This extensive fibrosis is indeed an interesting finding which was discussed in the first revision. In the first revision the reviewers commented the following:

Development of fibrosis is a hallmark of type 2 immune bias (e.g. liver fibrosis in Schistosoma infection). Given the abundance of M2-like macrophages in the aortic tissue in vascular Q fever, I wonder whether this potential link to elastin breakdown and fibrosis should not be made in the Discussion section.

Based on this comment, we have further improved the paragraph elaborating on fibrosis in the Discussion section to the following:

“The last major difference between Q fever AAA and atherosclerotic AAA is the extent of damage to the vascular wall architecture in Q fever AAAs. This is demonstrated by extensive loss of elastin fibers and increase of fibrosis present in the vascular wall. Fragmentation of elastin fibers has been described for AAAs previously (23), however we found the loss of elastin fibers more evident in the lesions from Q fever AAAs than atherosclerotic AAAs. Fibrosis is characterized by replacement of normal tissue by excessive connective tissue, and usually follows chronic inflammation. Presence of fibrosis is a sign of a type 2 immune response (42), which we also demonstrated in our cohort with the abundance of M2 macrophages. This may be the effect of persistent presence of growth factors, proteolytic enzymes, angiogenic factors, and profibrotic cytokines. (43, 44) Previously, fibrosis was also observed in chronic Q fever endocarditis in humans and cows. (12, 45-47) This indicates that our Q fever AAA cohort suffered from more destructive disease than our AAA cohort.”

In this paragraph we elucidate the presence of fibrosis in context of potential mechanisms and literature describing fibrosis in chronic Q fever endocarditis in both human and bovine tissue. We believe that this section contains all potential perspectives on fibrosis development in Q fever AAAs. Please inform us if we misinterpreted this comment, so we can improve our manuscript accordingly.